# Knowledge Distillation Detection
# for Open-weights Models

**Qin Shi**[*1]**, Amber Yijia Zheng**[*2]**, Qifan Song**[1]**, Raymond A. Yeh**[2]

[1]Department of Statistics, Purdue University
[2]Department of Computer Science, Purdue University
{shi622, zheng709, qfsong, rayyeh}@purdue.edu
[*]Equal contribution.

## Abstract

We propose the task of *knowledge distillation detection*, which aims to determine whether a student model has been distilled from a given teacher, under a practical setting where only the student's weights and the teacher's API are available. This problem is motivated by growing concerns about model provenance and unauthorized replication through distillation. To address this task, we introduce a model-agnostic framework that combines data-free input synthesis and statistical score computation for detecting distillation. Our approach is applicable to both classification and generative models. Experiments on diverse architectures for image classification and text-to-image generation show that our method improves detection accuracy over the strongest baselines by 59.6% on CIFAR-10, 71.2% on ImageNet, and 20.0% for text-to-image generation. The code is available at https://github.com/shqii1j/distillation_detection.

## 1 Introduction

Knowledge distillation [20] transfers learned knowledge from a larger teacher model to a smaller student model to reduce the computational and storage requirements. As opposed to directly training the smaller model from scratch alone, knowledge distillation leads to improved generalization of the student model and has found success in several areas, including image classification [8, 20, 35, 43, 63], large language models [16, 40, 49], and text-to-image generation [23, 28, 45, 56].

While the original intent of knowledge distillation is to build efficient models, distillation techniques lead to an unintended consequence that one could potentially clone proprietary teacher models without permission. This raises concerns about the violation of intellectual property and how to attribute the source of machine learning models [33, 41, 50]. In this work, we aim to study whether it is possible to *detect* if a model has been distilled from a given teacher model, in short, *knowledge distillation detection*. Specifically, we consider the realistic scenario of detecting distillation for open-weight student models, without making any assumptions about the distillation method, training data, or requiring the weights of the teacher model.

We also consider formulating the problem in a multiple-choice setting: given a student model and a set of candidate teachers, predict which teacher was used for distillation. This setting has two advantages for the ease of experiment setup. First, it avoids the need to calibrate an absolute threshold, as the decision is based on comparing across candidates, which is more common in real-world cases. Second, it produces results that are easier to interpret, as one can directly identify the most likely teacher. Note, the detection (binary) setting can be recovered from this formulation by applying a calibrated score function and threshold.

39th Conference on Neural Information Processing Systems (NeurIPS 2025).

Prior works have considered related problems such as membership inference [5, 22] and out-of-distribution detection [29]. These works demonstrate that models may retain statistical traces of their training data or source model, even after distillation or fine-tuning. In particular, recent work on auditing data usage [22] and memorization in generative models [6, 47] shows that it is possible to identify some aspects of model origin through probing-based methods. However, these approaches typically focus on detecting training data membership, rely on access to the teacher model, or assume architectural similarity. They **do not** directly tackle the problem of determining whether a model is distilled from another model that we are interested in.

To address knowledge distillation detection, we propose a model-agnostic approach that only requires access to the student model's weights, without requiring the teacher model's weights or training data. Our approach consists of three stages: input construction, score computation, and decision making. In the first stage, we generate synthetic queries that probe model behavior, using data-free synthesis techniques tailored to the model modality. In the second stage, we extract statistical scores from the model's responses, capturing either point-wise discrepancies or set-level distributional alignment. Finally, we apply a decision rule to determine whether the model has been distilled, based on these scores. This design is general and thus supports a wide range of models and tasks.

We conduct extensive experiments on both image classification and text-to-image generation tasks. For classification, we test a variety of architectures (ResNet18, DLA, DPN, DenseNet, ResNet50, and ResNeXt50) and distillation methods (KD, RKD, OFA-KD). For text-to-image generation, we report on student models distilled from SD-v2.1, SDXL, and PixArt, including BK-SDM, AMD, DMD2, and SDXL-Lightning. Across both domains, our method improves detection accuracy over the strongest baselines by 59.6% on CIFAR-10, 71.2% on ImageNet, and 20.0% for text-to-image generation.

**Our contributions are summarized as follows:**

- We introduce the task of knowledge distillation detection of open-weights student models, without access to the teacher model or training data.
- We propose a model-agnostic approach consisting of input construction, score computation, and decision rule stages, adaptable across both classification and generative settings.
- Empirical results on both classification and text-to-image tasks shows the effectiveness and generality of our method.

## 2   Related Work

**Knowledge distillation (KD)** is an effective technique for compressing information from large pre-trained models into lightweight student models by aligning either the outputs or intermediate representations of the teacher and student [8, 37, 43, 48, 61, 63, 68]. The concept was first introduced by Buciluă et al. [3] and later popularized by Hinton et al. [20]. For classification tasks, knowledge distillation methods are commonly grouped into three categories: logit-based [20], feature-based [18, 43], and relation-based approaches [10, 35].

While classification-focused KD emphasizes improving accuracy or reducing model size, distillation for diffusion models primarily aims to accelerate generation [23, 28, 31, 32, 42, 44–46, 53, 56]. A typical approach is to train a student model to approximate the teacher's sampling trajectory, often formulated as an ODE, with significantly fewer inference steps. For instance, AMD [45] distills generative features from the teacher; BK-SDM [23] uses block pruning and feature distillation; and DMD [57] matches the output distribution directly via KL divergence without step-wise supervision. Unlike prior work that focuses on improving the performance of distillation, we study the inverse problem to detect whether one model is distilled from the other.

**Data-free quantization and distillation.** We review two different tasks that use sample synthesis to train a quantized/student model. One task is data-free model quantization [52], which targets to quantize weights, activations, and even gradients to low-precision, to yield highly compact models without access to the training data. Data-free knowledge distillation [30] is a technique in which a smaller student model is trained to replicate the behavior of a teacher model without access to the original training data. Both tasks share the goal of transferring knowledge from a pre-trained teacher model to a target model using synthetic data. Common approaches include directly learning a fixed set of representative images [4, 27] or training a generator to produce synthetic

inputs [2, 11, 38, 52, 55, 59]. In our work, we also synthesize inputs as we do not assume access to the training data; however, our focused task is different. Rather than aiming for a quantized or distilled model, we generate these samples to collect statistics from the student/teacher model to perform knowledge distillation detection.

**AI for good.** Open-weight models can be freely shared and modified, amplifying both benefits and risks, which makes safety, transparency, and accountability crucial. Prior work on model immunization [64–67] develops defenses that reduce vulnerabilities of open-sourced image classification and diffusion models to harmful adaptations. Other efforts focus on safety interventions for large language models, such as red-teaming and watermarking [13, 24, 36], to mitigate misuse and enable provenance tracking. Our work is complementary: instead of altering model behavior, we address accountability by detecting whether a student has been distilled from a teacher, providing a mechanism for provenance and trust in open-weights generative models.

## 3 Background

**Knowledge distillation.** Let $f : \mathcal{X} \to \mathcal{Y}$ be the teacher model and $g_\theta : \mathcal{X} \to \mathcal{Y}$ be the student model, where $\mathcal{X}$ and $\mathcal{Y}$ are the input and output spaces. Given a training dataset $\mathcal{D} = \{(\boldsymbol{x}, \boldsymbol{y})\}$, knowledge distillation trains the student model by considering both the ground-truth supervision and the knowledge from the teacher model. The training objective can be formulated as

$$\mathcal{L}_{\text{KD}}(\theta) = \sum_{(\boldsymbol{x}, \boldsymbol{y}) \in \mathcal{D}} (1 - \lambda) \cdot \ell_{\text{hard}}(g_\theta(\boldsymbol{x}), \boldsymbol{y}) + \lambda \cdot \ell_{\text{soft}}(g_\theta(\boldsymbol{x}), f(\boldsymbol{x})), \tag{1}$$

where $\lambda \in [0, 1]$ balances the loss term, $\ell_{\text{hard}}$ denotes the loss with respect to the true label $\boldsymbol{y}$, and $\ell_{\text{soft}}$ compares the student's and teacher's outputs, *e.g.*, using KL divergence over softened logits.

## 4 Knowledge Distillation Detection

**Task formulation.** Given an open-weight distilled student model $g_\theta$, the goal of distillation detection is to identify which teacher model, from a set $\mathcal{F} \triangleq \{f^{(1)}, f^{(2)}, \ldots, f^{(K)}\}$, was used to distill the student. For a realistic setting, we assume the following at prediction time: (a) that there is API access to each of the teacher models, (b) the student model is open-weight, (c) the knowledge distillation algorithm is unknown, and (d) the data used to train or distill the models are unavailable.

More formally, this task can be formulated as a multiple-choice problem for the $K$ teacher candidates, *i.e.*, we aim to develop a detection algorithm

$$\mathcal{A} : \{g_\theta : \mathcal{X} \to \mathcal{Y}\} \to \{1, \ldots, K\} \tag{2}$$

that maps from a given student model to the index of the teacher model. That is, if the algorithm returns $k$, then it is most likely that the teacher $f_k$ is used to distill $g_\theta$. Compared with the task of deciding whether a student is distilled from a specific teacher, the multiple-choice setup is easier to interpret. It compares the student against a set of candidate teachers and identifies the most likely source without requiring an explicit threshold; see Sec. 4.1. Moreover, our method naturally extends to the binary setting: one can test whether a student is distilled from a given teacher by applying a threshold to the score. Results of this pairwise detection are discussed in Sec. 5.3.

### 4.1 Approach

At a high level, determining whether a student model has been distilled from a teacher model involves comparing their outputs. In our setting, however, no data is available; thus, the detection algorithm must first construct synthetic inputs. Once such inputs are obtained, both the teacher and student models are used to produce outputs, which are then compared to compute point-wise scores. These scores are then aggregated to make a final prediction. We illustrate our approach in Fig. 1. As the method's implementation may vary depending on the task, we first present the general framework, followed by task-specific choices in Sec. 4.2 and Sec. 4.3.

**Prediction.** Our proposed detection algorithm $\mathcal{A}$ is based on score maximization to identify the teacher model most likely used for distillation. Specifically, the predicted teacher index is given by

$$k^* = \underset{k \in \{1, \ldots, K\}}{\arg\max} \ S(g_\theta, f^{(k)}, \mathcal{P}), \tag{3}$$

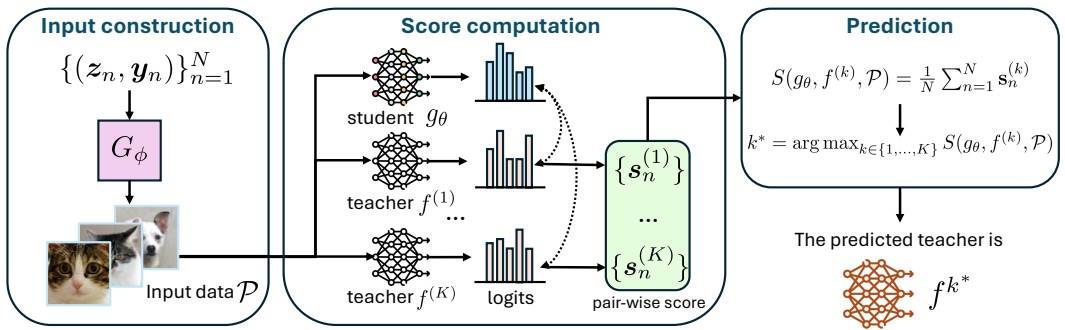

Figure 1: **Knowledge distillation detection pipeline.** The framework consists of three stages: input construction via a generator $G$, score computation between the student and candidate teachers, and prediction by selecting the teacher with the highest aggregated score.

where $S$ denotes a general score function that quantifies the "alignment" between the student model $g_\theta$ and the $k^{\text{th}}$ teacher model $f^{(k)}$, evaluated over a constructed input set $\mathcal{P} = \{x_n\}_{n=1}^N$.

**Score computation.** The score $S$ plays a central role in our method, which reflects how well the student aligns with each candidate teacher. We consider two types of score functions that capture this alignment at different levels:

• *Point-wise score* $S_{\text{point}}$ computes the discrepancy between the student and teacher outputs independently on each input $x_n \in \mathcal{P}$. Let $\delta : \mathcal{Y} \times \mathcal{Y} \to \mathbb{R}_{\geq 0}$ denote a general distance or divergence measure, *e.g.*, KL divergence. For each input $x_n$, we compute a pairwise score $s_n^{(k)}$ between the student and the $k^{\text{th}}$ teacher defined as:

$$s_n^{(k)} = \frac{1}{\delta\left(g_\theta(x_n), f^{(k)}(x_n)\right) + \epsilon}, \tag{4}$$

where $\epsilon > 0$ is a small constant to ensure numerical stability. The overall point-wise score is then given by the average over all $N$ constructed inputs:

$$S_{\text{point}}(g_\theta, f^{(k)}, \mathcal{P}) = \frac{1}{N} \sum_{n=1}^N s_n^{(k)}. \tag{5}$$

• *Set-level score* $S_{\text{set}}$ measure the *population* discrepancy between $g_\theta$ and each $f^{(k)}$ by comparing their outputs across the entire input set $\mathcal{P}$. The score takes on the following form:

$$S_{\text{set}}(g_\theta, f^{(k)}, \mathcal{P}) = \Gamma\left(\left\{\left(g_\theta(x_n), f^{(k)}(x_n)\right)\right\}_{x_n \in \mathcal{P}}\right), \tag{6}$$

where $\Gamma : (\mathcal{Y} \times \mathcal{Y})^N \to \mathbb{R}$ is a function that operates on the joint set of outputs to capture alignment patterns over the set, such as closeness in the distribution.

**Input construction.** As discussed, a set of inputs $\mathcal{P} = \{x_n\}_{n=1}^N$ is required to collect output statistics from both the teacher and student models. Intuitively, the method would benefit from inputs $x_n$ that yield scores which are "separable", *i.e.*, they can distinguish the true teacher model used for distillation from other candidate teachers. For example, inputs that are overfitted by both the teacher and student models may be a strong indicator of a distillation relationship. Hence, our input construction strategy is designed to favor inputs with confident predictions from the student model, which tend to improve the discriminability of the subsequent computed scores.

We will next describe the task-specific details for detecting distillation in image classification and text-to-image generation.

### 4.2 Image Classification Specific Details

In the classification setting, both the teachers $\{f^{(k)}\}$ and the student $g_\theta$ are multi-class predictors, *i.e.*, $\mathcal{X} \to \Delta^C$, where $\Delta^C$ denotes the probability simplex over $C$ classes.

For *point-wise score*, we choose the pairwise score $s_n^{(k)}$ to be computed using the KL divergence between the teacher and student's output. For a generated input $x$, we compute the discrepancy as $D_{\mathrm{KL}}\left(g_\theta(x)\|f^{(k)}(x)\right)$, and convert it into an aggregated point-wise score $S_{\mathrm{point}}$ using the inverse formulation in Eq. (5).

For *set-level score*, we choose $\Gamma$ to be Aligned Cosine Similarity (ACS) [17], which computes the average cosine similarity between linearly projected student and teacher logits across a set of inputs:

$$\mathrm{ACS}(\boldsymbol{G}, \boldsymbol{F}) = \frac{1}{N} \sum_{n=1}^{N} \frac{\langle (H\boldsymbol{G})_{n\cdot}, (H\boldsymbol{F}R)_{n\cdot} \rangle}{\|(H\boldsymbol{G})_{n\cdot}\|_2 \|(H\boldsymbol{F}R)_{n\cdot}\|_2}, \tag{7}$$

where $\boldsymbol{G} = [g_\theta(\boldsymbol{x}_1), \dots, g_\theta(\boldsymbol{x}_N)]^\top$ and $\boldsymbol{F} = [f^{(k)}(\boldsymbol{x}_1), \dots, f^{(k)}(\boldsymbol{x}_N)]^\top$ are matrices of student and teacher logits, $H = I_N - \frac{1}{N}\mathbf{1}_N\mathbf{1}_N^\top$ is the centering function and

$$R = \underset{R \in \{R^\top R = I\}}{\arg\min} \|H\boldsymbol{G} - H\boldsymbol{F}R\|_F \tag{8}$$

is a projector for mapping centered $\boldsymbol{F}$ to centered $\boldsymbol{G}$. Here, we use an orthogonal transformation obtained by performing singular value decomposition (SVD). This set-level score captures distribution-level alignment between the two models in a compact and representation-aware manner.

To construct the input set $\mathcal{P}$, we aim to generate in-distribution inputs. We adopt a mixup-based synthesis strategy [11, 60], where each generated input is a label-conditioned combination of latent representations from a generator $G_\phi$, *i.e.*,

$$\hat{\boldsymbol{x}} = G_\phi\left(\sum_{i=1}^{C} \boldsymbol{w}_i \cdot e(\boldsymbol{z}_i, \boldsymbol{y}_i)\right). \tag{9}$$

Here, the generator $G_\phi$ is parametrized with a deep net, with $\{\boldsymbol{z}_i\}_{i=1}^C \sim \mathcal{N}(0, I)$, labels $\{\boldsymbol{y}_i\}_{i=1}^C \subset \mathcal{Y}$, interpolation weights $\{\boldsymbol{w}_i\}_{i=1}^C \sim \mathrm{Dir}(\mathbf{1})$ for mixup, and $e(\boldsymbol{z}_i, \boldsymbol{y}_i)$ denoting one label-conditioned latent encoding, which is a linear layer that takes in the concatenation of $\boldsymbol{z}_i$ and $\boldsymbol{y}_i$.

The generator $G_\phi$ is optimized to produce data $\hat{\boldsymbol{x}}$ that has high confidence from $g_\theta$ as follows,

$$\min_\phi \sum_{i=1}^{C} \boldsymbol{w}_i \cdot \mathcal{L}_{\mathrm{hard}}(g_\theta(\hat{\boldsymbol{x}}(\phi)), \boldsymbol{y}_i) + \mathcal{L}_{\mathrm{BNS}}(g_\theta, \hat{\boldsymbol{x}}(\phi)), \tag{10}$$

where we use the cross-entropy loss for $\mathcal{L}_{\mathrm{hard}}$, and $\mathcal{L}_{\mathrm{BNS}}$ is the batch normalization statistics (BNS) alignment loss [55]. Specifically, the BNS loss encourages the generated samples to match the internal activation statistics of real data by aligning the mean and variance at each batch norm layer:

$$\mathcal{L}_{\mathrm{BNS}}(G_\phi, g_\theta) = \sum_{l=1}^{L} \|\boldsymbol{\mu}_l^r - \boldsymbol{\mu}_l\|_2^2 + \|\boldsymbol{\sigma}_l^r - \boldsymbol{\sigma}_l\|_2^2, \tag{11}$$

where $\boldsymbol{\mu}_l^r$ and $\boldsymbol{\sigma}_l^r$ are the batch mean and variance of the generated data from $G_\phi$ at the $l$-th batch norm layer, and $\boldsymbol{\mu}_l, \boldsymbol{\sigma}_l$ are the corresponding running statistics stored in the pre-trained model $g_\theta$.

After the generator $G_\phi$ is trained, we generate class-conditional synthetic samples for each label $\boldsymbol{y}$ without mixup by sampling new noise $\boldsymbol{z} \sim \mathcal{N}(0, I)$ to construct the input set

$$\mathcal{P} = \left\{ G_\phi\left(e(\boldsymbol{z}, \boldsymbol{y})\right) \middle| \boldsymbol{z} \sim \mathcal{N}(0, 1) \, \forall \boldsymbol{y} \right\}. \tag{12}$$

### 4.3 Text-to-image Generation Specific Details

In the text-to-image setting, both the teachers $\{f^{(k)}\}$ and the student model $g_\theta$ are conditional generative models that map a text prompt to an image, *i.e.*, $\mathcal{X} \to \mathcal{I}$, where $\mathcal{X}$ is the text input space and $\mathcal{I}$ is the image space.

For *point-wise score* $S_{\mathrm{point}}$, we choose to compute pairwise score $s_n^{(k)}$ using LPIPS [62], which measures the semantic similarity between images generated by the student and teacher models for the same prompt and the same noise latent.

For the *set-level score* $S_{\text{set}}$, we choose to apply Centered Kernel Alignment (CKA) [25] with RBF kernels as a similarity index to measure the global alignment between student and teacher models. Concretely, we use a pre-trained CLIP image encoder [39] to extract representation embeddings from images generated by both the student and teacher models, and apply CKA to compare the two distributions of embeddings. Compared to the ACS used for the classification task, the CKA is flexible to complex distributions and robust to high dimensionality, thus it is more suitable for the text-to-image task.

Given two sets of image embeddings $\boldsymbol{X}$ and $\boldsymbol{Y}$, each extracted from images generated from the student and teacher models, we first compute the RBF kernel matrices:

$$K_{\boldsymbol{X}} = \exp\left(-\frac{1}{2\sigma^2}D_{\boldsymbol{X}}\right), \quad D_{\boldsymbol{X}} = \text{diag}(\boldsymbol{X}\boldsymbol{X}^\top)\mathbf{1}^\top + \mathbf{1}\,\text{diag}(\boldsymbol{X}\boldsymbol{X}^\top)^\top - 2\boldsymbol{X}\boldsymbol{X}^\top, \quad (13)$$

and similar for $K_{\boldsymbol{Y}}$. With the centering function $H = I_n - \frac{1}{n}\mathbf{1}_n\mathbf{1}_n^\top$, the CKA score is defined as

$$\text{CKA}(\boldsymbol{X}, \boldsymbol{Y}) = \frac{\text{Tr}(\tilde{K}_{\boldsymbol{X}}\tilde{K}_{\boldsymbol{Y}})}{\|\tilde{K}_{\boldsymbol{X}}\|F \cdot \|\tilde{K}_{\boldsymbol{Y}}\|F} \quad \text{where } \tilde{K}_{\boldsymbol{X}} = HK_{\boldsymbol{X}}, \ \tilde{K}_{\boldsymbol{Y}} = HK_{\boldsymbol{Y}}. \quad (14)$$

Lastly, to generate the set of outputs from the teacher and student models, we need an input set $\mathcal{P}$. In the case of a text-to-image model, the input requires text prompts to specify what to generate. Here, we simply use the empty string as the prompt. This choice ensures that the input remains in distribution to the model trained for classifier-free guidance The training involves randomly omitting (replacing with an empty string) the text condition, usually with a drop rate of 0.05, to enable unconditional generation. As a result, the model learns to treat empty prompts as valid inputs, which makes them stable and suitable for evaluating the model's behavior in distillation detection.

# 5 Experiments

To evaluate the effectiveness and generalizability of our approach, we conduct experiments on detecting knowledge distillation in two tasks: image classification and text-to-image generation.

**Evaluation metrics.** As knowledge distillation is formulated as a multi-class classification problem, we consider the following metrics:

1. *Accuracy (Acc.):* For all student models, we compute the proportion for which the predicted teacher matches the true teacher. That is, we report how often our method correctly identifies the teacher model used for distillation.

2. *Area Under the Curve (AUC):* For each student model, we view the prediction as a one-vs-rest binary classification by treating the true teacher as the positive class and all others as negative. We compute the area under the ROC curve and report the average AUC across all student models. This metric reflects how well the scores rank the true teacher higher than the incorrect ones.

## 5.1 Distillation Detection on Image Classification

**Datasets and model architectures.** For image classification, we use two standard datasets: CIFAR-10 and ImageNet [14]. On CIFAR-10, we select four popular network architectures: ResNet-18 [19], DLA [58], DPN-92 [9], and DenseNet-121 [21]. For each architecture, we first train a teacher model from scratch. Then, for each teacher model, we distill its knowledge into student models of all four architectures, resulting in 16 student models for each knowledge distillation method.

On ImageNet, we consider three architectures: ResNet-18 [19], ResNet-50 [19], and ResNeXt-50 [51]. For each architecture, we use a teacher model pre-trained on ImageNet-1K and distill student models on ImageNet-100 into all three architectures, resulting in 9 student models for each knowledge distillation method. If the logit-matching is required during distillation, we only match the logits corresponding to the ImageNet-100 labels.

**Model distillation methods.** In image classification tasks, we consider the original knowledge distillation (KD) [20], rational knowledge distillation (RKD) [35], and one-for-all KD (OFA) [18]. Each student model is distilled using one of these approaches from a given teacher model.

**Baselines.** We explore several different baselines following the general framework of our approach, drawing inspiration from the scores used in related works, such as membership inference attacks,

Table 1: Detecting distillation on CIFAR-10. Each cell reports the mean and standard deviation of Acc./AUC over 10 random seeds. Bold indicates the highest Accuracy or AUC in each column (excluding Oracle). Note that when $N = 1$, the set-level scores cannot be computed.

| Method | Input Size $N$ | | | | | Average |
|---|---|---|---|---|---|---|
| | 1 | 5 | 10 | 50 | 100 | |
| Oracle | $0.45_{\pm 0.10}/0.64_{\pm 0.07}$ | $0.58_{\pm 0.10}/0.77_{\pm 0.06}$ | $0.65_{\pm 0.07}/0.82_{\pm 0.06}$ | $0.87_{\pm 0.04}/0.96_{\pm 0.01}$ | $0.95_{\pm 0.02}/0.99_{\pm 0.01}$ | $0.70_{\pm 0.07}/0.84_{\pm 0.04}$ |
| MIA Filter + KL | $0.43_{\pm 0.06}/0.66_{\pm 0.04}$ | $0.49_{\pm 0.05}/0.74_{\pm 0.02}$ | $0.51_{\pm 0.04}/0.76_{\pm 0.02}$ | $0.54_{\pm 0.03}/0.79_{\pm 0.01}$ | $0.55_{\pm 0.02}/0.80_{\pm 0.01}$ | $0.51_{\pm 0.04}/0.75_{\pm 0.02}$ |
| OOD Filter + KL | $0.42_{\pm 0.06}/0.68_{\pm 0.06}$ | $0.48_{\pm 0.03}/0.73_{\pm 0.03}$ | $0.50_{\pm 0.04}/0.75_{\pm 0.03}$ | $0.55_{\pm 0.03}/0.79_{\pm 0.01}$ | $0.54_{\pm 0.03}/0.80_{\pm 0.01}$ | $0.50_{\pm 0.04}/0.75_{\pm 0.03}$ |
| MIA Filter + CKA | - | $0.32_{\pm 0.05}/0.58_{\pm 0.05}$ | $0.38_{\pm 0.11}/0.64_{\pm 0.06}$ | $0.64_{\pm 0.06}/0.85_{\pm 0.04}$ | $0.72_{\pm 0.06}/0.90_{\pm 0.04}$ | $0.52_{\pm 0.07}/0.74_{\pm 0.05}$ |
| OOD Filter + CKA | - | $0.33_{\pm 0.09}/0.59_{\pm 0.06}$ | $0.39_{\pm 0.06}/0.65_{\pm 0.06}$ | $0.61_{\pm 0.08}/0.84_{\pm 0.04}$ | $0.72_{\pm 0.05}/0.91_{\pm 0.02}$ | $0.51_{\pm 0.07}/0.75_{\pm 0.05}$ |
| MIA Filter + MMD | - | $0.40_{\pm 0.08}/0.67_{\pm 0.04}$ | $0.45_{\pm 0.06}/0.70_{\pm 0.04}$ | $0.36_{\pm 0.03}/0.62_{\pm 0.03}$ | $0.29_{\pm 0.02}/0.58_{\pm 0.02}$ | $0.38_{\pm 0.04}/0.64_{\pm 0.03}$ |
| OOD Filter + MMD | - | $0.47_{\pm 0.07}/0.70_{\pm 0.03}$ | $0.47_{\pm 0.05}/0.71_{\pm 0.05}$ | $0.38_{\pm 0.03}/0.65_{\pm 0.04}$ | $0.28_{\pm 0.02}/0.59_{\pm 0.02}$ | $0.40_{\pm 0.04}/0.66_{\pm 0.03}$ |
| **Ours (KL)** | $\mathbf{0.62}_{\pm 0.09}/\mathbf{0.75}_{\pm 0.06}$ | $\mathbf{0.82}_{\pm 0.05}/\mathbf{0.89}_{\pm 0.04}$ | $\mathbf{0.86}_{\pm 0.04}/\mathbf{0.92}_{\pm 0.02}$ | $0.87_{\pm 0.03}/\mathbf{0.94}_{\pm 0.01}$ | $0.87_{\pm 0.02}/\mathbf{0.94}_{\pm 0.01}$ | $0.81_{\pm 0.05}/\mathbf{0.89}_{\pm 0.03}$ |
| **Ours (ACS)** | - | $0.75_{\pm 0.04}/0.72_{\pm 0.01}$ | $0.82_{\pm 0.04}/0.74_{\pm 0.01}$ | $\mathbf{0.88}_{\pm 0.03}/0.77_{\pm 0.01}$ | $\mathbf{0.89}_{\pm 0.04}/0.77_{\pm 0.01}$ | $\mathbf{0.83}_{\pm 0.04}/0.75_{\pm 0.01}$ |

Table 2: Detecting distillation on ImageNet. Each cell reports Acc./AUC. Bold indicates the best value per column (excluding Oracle).

| Method | Input Size $N$ | | | | | Average |
|---|---|---|---|---|---|---|
| | 1 | 5 | 10 | 50 | 100 | |
| Oracle | $0.56_{\pm 0.22}/0.71_{\pm 0.22}$ | $0.67_{\pm 0.16}/0.88_{\pm 0.14}$ | $0.77_{\pm 0.15}/0.94_{\pm 0.06}$ | $0.84_{\pm 0.12}/0.99_{\pm 0.02}$ | $0.91_{\pm 0.08}/1.00_{\pm 0.00}$ | $0.75_{\pm 0.15}/0.90_{\pm 0.09}$ |
| MIA Filter + KL | $0.34_{\pm 0.03}/0.75_{\pm 0.05}$ | $0.33_{\pm 0.01}/0.76_{\pm 0.03}$ | $0.33_{\pm 0.00}/0.76_{\pm 0.02}$ | $0.33_{\pm 0.00}/0.76_{\pm 0.01}$ | $0.33_{\pm 0.00}/0.76_{\pm 0.01}$ | $0.33_{\pm 0.01}/0.76_{\pm 0.03}$ |
| OOD Filter + KL | $0.34_{\pm 0.05}/0.62_{\pm 0.06}$ | $0.33_{\pm 0.03}/0.62_{\pm 0.02}$ | $0.32_{\pm 0.03}/0.62_{\pm 0.03}$ | $0.32_{\pm 0.02}/0.63_{\pm 0.01}$ | $0.31_{\pm 0.02}/0.63_{\pm 0.01}$ | $0.33_{\pm 0.03}/0.62_{\pm 0.03}$ |
| MIA Filter + CKA | - | $0.41_{\pm 0.06}/0.56_{\pm 0.05}$ | $0.33_{\pm 0.06}/0.51_{\pm 0.06}$ | $0.38_{\pm 0.08}/0.54_{\pm 0.09}$ | $0.42_{\pm 0.10}/0.55_{\pm 0.08}$ | $0.39_{\pm 0.07}/0.54_{\pm 0.07}$ |
| OOD Filter + CKA | - | $0.36_{\pm 0.09}/0.55_{\pm 0.10}$ | $0.34_{\pm 0.11}/0.48_{\pm 0.09}$ | $0.36_{\pm 0.09}/0.49_{\pm 0.12}$ | $0.36_{\pm 0.07}/0.57_{\pm 0.05}$ | $0.36_{\pm 0.09}/0.52_{\pm 0.09}$ |
| MIA Filter + MMD | - | $0.34_{\pm 0.01}/0.50_{\pm 0.04}$ | $0.33_{\pm 0.00}/0.50_{\pm 0.00}$ | $0.33_{\pm 0.00}/0.50_{\pm 0.00}$ | $0.33_{\pm 0.00}/0.50_{\pm 0.00}$ | $0.33_{\pm 0.00}/0.50_{\pm 0.01}$ |
| OOD Filter + MMD | - | $0.34_{\pm 0.01}/0.47_{\pm 0.04}$ | $0.33_{\pm 0.00}/0.50_{\pm 0.00}$ | $0.33_{\pm 0.00}/0.50_{\pm 0.00}$ | $0.33_{\pm 0.00}/0.50_{\pm 0.00}$ | $0.33_{\pm 0.00}/0.49_{\pm 0.01}$ |
| **Ours (KL)** | $\mathbf{0.47}_{\pm 0.13}/\mathbf{0.68}_{\pm 0.13}$ | $\mathbf{0.64}_{\pm 0.09}/\mathbf{0.88}_{\pm 0.04}$ | $\mathbf{0.66}_{\pm 0.07}/\mathbf{0.89}_{\pm 0.04}$ | $0.74_{\pm 0.04}/\mathbf{0.93}_{\pm 0.01}$ | $0.75_{\pm 0.03}/\mathbf{0.92}_{\pm 0.01}$ | $0.65_{\pm 0.07}/\mathbf{0.86}_{\pm 0.05}$ |
| **Ours (ACS)** | - | $0.58_{\pm 0.08}/0.66_{\pm 0.07}$ | $0.60_{\pm 0.09}/0.66_{\pm 0.04}$ | $0.70_{\pm 0.07}/0.81_{\pm 0.03}$ | $\mathbf{0.79}_{\pm 0.02}/0.85_{\pm 0.02}$ | $\mathbf{0.67}_{\pm 0.06}/0.74_{\pm 0.04}$ |

although our application differs. Each baseline alters one component of our method while keeping the other elements unchanged. Baselines 1-2 focus on the input construction process, while baselines 3-4 concentrate on the score design. Please see the Appendix for more details.

1. *Membership inference attack filter (MIA filter)* [12, 54]: This baseline constructs the input set by identifying samples that resemble training data. We use random noise as input and split the samples into two disjoint subsets. For the first subset, we collect the student model's output logits and randomly assign `Train` and `Test` labels to them. A binary classifier is then trained to distinguish between the two. On the second subset, we collect logits from the student model and pass them through the trained classifier. Inputs classified as `Train` are retained for detection, while those classified as `Test` are discarded. KL divergence is used as the score function in the subsequent stage.

2. *Out-of-distribution detector filter (OOD filter)* [29]: This baseline constructs the input set by filtering for in-distribution samples. It computes an energy score from the model's logits for random noise inputs. Samples with low energy (high confidence) are treated as in-distribution and retained, while high-energy (low-confidence) samples are filtered out. As in our method, the KL divergence is used for scoring.

3. *MMD-FUSE* [1]: It computes the normalized log-sum-exp of Maximum Mean Discrepancy (MMD) statistics between student and teacher logits outputs under permutations.

4. *CKA* [25]: It computes the representational similarity via centered kernel alignment. RBF kernels are applied to the logits of the student and teacher, followed by centering and computation of the Hilbert-Schmidt Independence Criterion (HSIC).

Besides the baselines above, we also introduce **Oracle**, where we directly sample real inputs from the training distribution in the input construction stage and use KL divergence as the score function. This setup illustrates the performance when the original training data is available.

**Results.** In Tab. 1, we present the results of distillation detection on CIFAR-10 across varying numbers of synthesized inputs $N = |\mathcal{P}|$. Set-level scores cannot be computed when $N = 1$. For point-level score, even with a single input, our method using KL divergence achieves strong performance, with an accuracy of 0.62 and an AUC of 0.75, consistently outperforming all baselines.

As $N$ increases, performance improves for both our methods and the baselines, reflecting the benefit of aggregating more input samples. By $N = 50$, our KL and ACS variants reach AUCs of 0.94 and 0.77, respectively, and accuracy continues to improve. In contrast, baselines tend to plateau around

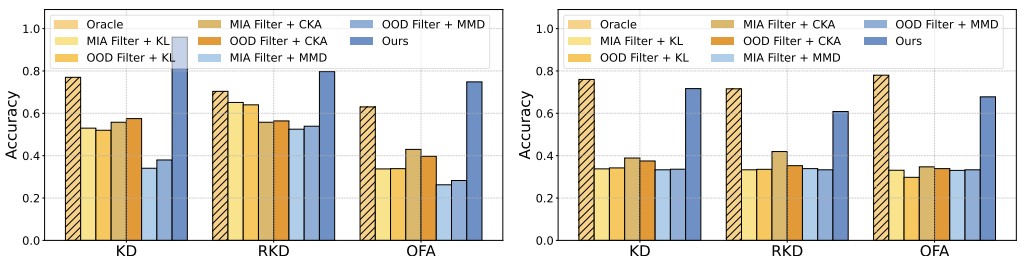

Figure 2: Accuracy of knowledge distillation detection across different distillation methods. CIFAR-10 (left) and ImageNet (right).

0.50–0.55 in accuracy and 0.73–0.80 in AUC. Notably, our methods exhibit greater stability and higher average performance across all settings, even outperforming the Oracle that uses real data.

Tab. 2 presents distillation detection results on ImageNet. As we can observe, our methods consistently outperform the baselines. Notably, with just 5 inputs, our KL-based method achieves an accuracy of 0.64 and AUC of 0.88, which is significantly higher than all baseline combinations, whose accuracy remains around 0.33 and AUC ranges from 0.47 to 0.76. As the number of inputs increases, our performance remains stable and continues to improve, reaching 0.75 accuracy and 0.92 AUC with 100 inputs. On average, our KL variant achieves 0.65/0.86, and ACS achieves 0.67/0.74, both substantially higher than the baselines, which mostly remain around 0.33–0.36 in accuracy.

In Fig. 2, we present the accuracy of detection on CIFAR10 and ImageNet across different distillation methods. Our method consistently achieves the highest accuracy across all distillation types, outperforming all baselines and even the Oracle in CIFAR-10. In addition, our method achieves the best performance under the standard KD setup compared to RKD and OFAKD, suggesting that detection is more effective when the student model is trained using vanilla knowledge distillation.

## 5.2 Detect Distillation on Text-to-image Models

**Model distillation methods and architectures.** For the text-to-image models, we use 10 publicly available pre-trained student models distilled from larger teacher models using the following:

1. AMD [45]: a one-step student model distilled from Stable Diffusion 2.1 and a one-step student model distilled from PixArt.
2. DMD2 [56]: four-step distilled models derived from Stable Diffusion XL (SDXL).
3. SDXL-Lightning [28]: student models with 1, 2, 4, and 8 sampling steps, all distilled from Stable Diffusion XL (SDXL).
4. BK-SDM-v2 [23]: improved variants of BK-SDM (base, small, and tiny), distilled from Stable Diffusion 2.1.

**Baselines.** For detection on text-to-image models, we consider two categories of baselines: captioning-based input construction stage and alignment-based set-level scores. Baseline 1-2 focuses on comparing the input construction process, and 3-6 focuses on the score design.

1. *BLIP-Base* [26]: We use the BLIP-Base model to generate captions for the unconditional generation of the student model. These captions are then used as input for the detection pipeline.
2. *GPT-2* [34]: Similar to BLIP-Base, we use the ViT-GPT2 model to caption for the unconditional generation of the student model, and feed the resulting text for detection.
3. *MMD-FUSE* [25]: As in the classification setting, we calculate $p$-value as the score to measure the alignment between the student and teacher. Here, it is applied to the generated images instead of the logits.
4. *ACS* [17]: This baseline computes the average cosine similarity between the projected logits of the student and teacher across a set of generated samples.
5. *CLIP* [39]: We compute the cosine similarity between the CLIP image embeddings of student and teacher generations.
6. *DINO* [7]: We extract features from generated images using a pre-trained DINO backbone and compute the cosine similarity between the resulting embeddings of the student and teacher.

Table 3: Detecting distillation in text-to-image generation model. Each cell reports Acc./AUC.

| Method | Input Size $N$ | | | | | Average |
|---|---|---|---|---|---|---|
| | 1 | 5 | 10 | 50 | 100 | |
| Blip-Base + CLIP | $0.71_{\pm0.10}$/$0.78_{\pm0.07}$ | $0.77_{\pm0.09}$/$0.91_{\pm0.04}$ | $0.81_{\pm0.08}$/$0.93_{\pm0.04}$ | $0.84_{\pm0.05}$/$0.96_{\pm0.01}$ | $0.83_{\pm0.05}$/$0.96_{\pm0.01}$ | $0.79_{\pm0.07}$/$0.91_{\pm0.03}$ |
| Blip-Base + DINO | $0.76_{\pm0.11}$/$0.83_{\pm0.06}$ | $0.79_{\pm0.09}$/$0.93_{\pm0.03}$ | $0.80_{\pm0.08}$/$0.94_{\pm0.03}$ | $0.80_{\pm0.04}$/$0.95_{\pm0.01}$ | $0.81_{\pm0.05}$/$0.95_{\pm0.01}$ | $0.79_{\pm0.08}$/$0.92_{\pm0.03}$ |
| Blip-Base + ACS | - | $0.56_{\pm0.08}$/$0.52_{\pm0.16}$ | $0.58_{\pm0.09}$/$0.63_{\pm0.17}$ | $0.59_{\pm0.07}$/$0.98_{\pm0.03}$ | $0.57_{\pm0.06}$/$1.00_{\pm0.00}$ | $0.57_{\pm0.08}$/$0.78_{\pm0.09}$ |
| Blip-Base + MMD | - | $0.67_{\pm0.10}$/$0.96_{\pm0.02}$ | $0.73_{\pm0.08}$/$0.97_{\pm0.01}$ | $0.58_{\pm0.06}$/$0.89_{\pm0.05}$ | $0.50_{\pm0.00}$/$0.67_{\pm0.02}$ | $0.62_{\pm0.06}$/$0.87_{\pm0.03}$ |
| GPT-2 + CLIP | $0.70_{\pm0.13}$/$0.81_{\pm0.09}$ | $0.73_{\pm0.06}$/$0.89_{\pm0.05}$ | $0.81_{\pm0.07}$/$0.93_{\pm0.02}$ | $0.81_{\pm0.03}$/$0.95_{\pm0.01}$ | $0.80_{\pm0.00}$/$0.95_{\pm0.01}$ | $0.77_{\pm0.06}$/$0.91_{\pm0.04}$ |
| GPT-2 + DINO | $0.81_{\pm0.13}$/$0.87_{\pm0.09}$ | $0.79_{\pm0.09}$/$0.91_{\pm0.04}$ | $0.80_{\pm0.06}$/$0.94_{\pm0.03}$ | $0.80_{\pm0.04}$/$0.95_{\pm0.01}$ | $0.80_{\pm0.00}$/$0.95_{\pm0.01}$ | $0.80_{\pm0.07}$/$0.92_{\pm0.04}$ |
| GPT-2 + ACS | - | $0.51_{\pm0.09}$/$0.47_{\pm0.17}$ | $0.57_{\pm0.11}$/$0.58_{\pm0.19}$ | $0.52_{\pm0.04}$/$0.97_{\pm0.04}$ | $0.54_{\pm0.05}$/$1.00_{\pm0.01}$ | $0.54_{\pm0.07}$/$0.75_{\pm0.10}$ |
| GPT-2 + MMD | - | $0.64_{\pm0.09}$/$0.95_{\pm0.01}$ | $0.68_{\pm0.06}$/$0.96_{\pm0.01}$ | $0.51_{\pm0.03}$/$0.89_{\pm0.03}$ | $0.50_{\pm0.00}$/$0.66_{\pm0.00}$ | $0.58_{\pm0.05}$/$0.87_{\pm0.01}$ |
| **Ours (LPIPS)** | $\mathbf{0.89}_{\pm0.05}$/$\mathbf{1.00}_{\pm0.00}$ | $\mathbf{0.94}_{\pm0.05}$/$\mathbf{1.00}_{\pm0.00}$ | $\mathbf{0.97}_{\pm0.05}$/$\mathbf{1.00}_{\pm0.00}$ | $\mathbf{1.00}_{\pm0.00}$/$\mathbf{1.00}_{\pm0.00}$ | $\mathbf{1.00}_{\pm0.00}$/$\mathbf{0.99}_{\pm0.01}$ | $\mathbf{0.96}_{\pm0.03}$/$\mathbf{1.00}_{\pm0.00}$ |
| **Ours (CKA)** | - | $0.67_{\pm0.23}$/$0.73_{\pm0.15}$ | $0.72_{\pm0.15}$/$0.83_{\pm0.17}$ | $0.78_{\pm0.09}$/$0.99_{\pm0.02}$ | $0.80_{\pm0.06}$/$1.00_{\pm0.00}$ | $0.74_{\pm0.13}$/$0.89_{\pm0.08}$ |

Table 4: Ablation study on input construction and score computation on classification.

| Setting | CIFAR-10 | ImageNet | Average |
|---|---|---|---|
| OOD filter + **ACS** | $0.56_{\pm0.05}$/$\mathbf{0.77}_{\pm0.04}$ | $0.37_{\pm0.07}$/$0.52_{\pm0.08}$ | $0.47_{\pm0.06}$/$0.65_{\pm0.06}$ |
| **Synthetic Data** + CKA | $0.82_{\pm0.03}$/$\mathbf{0.77}_{\pm0.01}$ | $0.62_{\pm0.07}$/$0.71_{\pm0.04}$ | $0.72_{\pm0.05}$/$0.74_{\pm0.03}$ |
| **Synthetic Data** + **ACS (Ours)** | $\mathbf{0.83}_{\pm0.04}$/$0.75_{\pm0.01}$ | $\mathbf{0.67}_{\pm0.06}$/$\mathbf{0.74}_{\pm0.04}$ | $\mathbf{0.75}_{\pm0.05}$/$\mathbf{0.75}_{\pm0.03}$ |

**Results.** Tab. 3 shows the performance of distillation detection for text-to-image generation models under varying numbers of synthesized inputs. Our methods consistently outperform all baselines in both accuracy and AUC across all input sizes, especially when we apply a point-wise score function. With just a single input, it achieves 0.89 accuracy and 1.00 AUC, already exceeding all competing methods. The strong performance is partly due to the use of CFG during distillation, where the unconditional generation is explicitly modeled, *i.e.*, the empty string is used in the distillation process.

Next, as the number of inputs increases, performance quickly saturates, reaching 1.00 accuracy and 1.00 AUC from $N = 10$ onward. This trend is maintained across all settings, with an average of 0.96/1.00, indicating strong robustness and effectiveness. The best-performing baselines, such as GPT-2 + DINO and Blip-Base + DINO, had around 0.80 accuracy, even with large input sizes.

For set-level scoring approaches, our method using CKA demonstrates clear improvements over other set-based baselines. Its accuracy and AUC increase steadily with $N$, reaching 0.80/0.99 at $N = 100$. In comparison, other baselines using set-level score ACS or MMD either stagnate or degrade when $N > 10$, indicating limited scalability. These results highlight the effectiveness of our design in leveraging larger input sets for improved detection.

## 5.3 Ablation and Discussion

**Pairwise detection.** To show that our multiple-choice setup can be extended to a binary detection, we implement the HSIC test [15] to obtain a p-value as the score for the text-to-image generation task with 3 teacher models and 10 student models. Each of the student models is paired with its true teacher (positive) and two random candidate teachers (negative). A pair is predicted as distillation if the p-value is below 0.05. As shown in Tab. 5, from the 30 teacher-student pairs, accuracy and F1 score are computed from these binary predictions, and AUC is based on the scores across different thresholds for p-value. This method can achieve 0.86 accuracy and 0.91 AUC, which shows that generalizing to the pairwise testing setup is possible. We view this as a natural extension and a valuable direction for future work.

**Ablate inputs and scores.** We ablate input construction and set-level scoring in Tab. 4 for classification, reporting both accuracy and AUC. Using OOD-filtered inputs with ACS yields the weakest results, especially on ImageNet with 0.37 accuracy and 0.52 AUC, indicating poor alignment with student model behavior. Replacing the input with our synthetic samples and using CKA improves performance significantly, reaching 0.72 accuracy and 0.74 AUC on average. Our full method

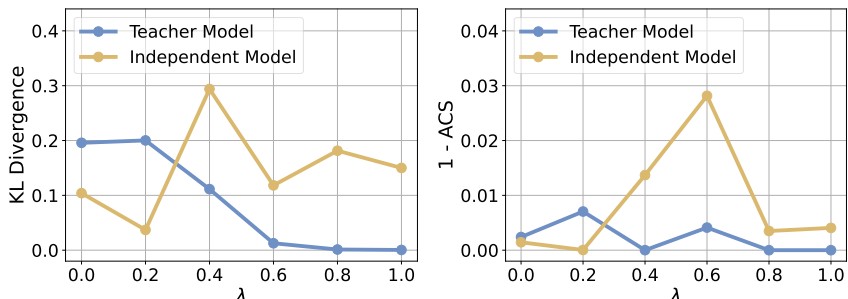

Figure 3: KL divergence (left) and 1 - ACS (right) between the student and teacher (or independent model) outputs as a function of the distillation weight $\lambda$. The total loss used to train the student is defined in Eq. (1), combining a hard cross-entropy term and a soft KL divergence term.

Table 5: Performance of distillation detection using the HSIC test on the text-to-image task.

| Input Size $N$ | Acc. | AUC | F1 |
|---|---|---|---|
| 10 | $0.74 \pm 0.05$ | $0.76 \pm 0.07$ | $0.43 \pm 0.09$ |
| 50 | $0.86 \pm 0.04$ | $0.91 \pm 0.07$ | $0.80 \pm 0.06$ |
| 100 | $0.80 \pm 0.04$ | $0.94 \pm 0.04$ | $0.75 \pm 0.05$ |

achieves the best overall results 0.75/0.75, outperforming the strongest ablation by 0.03 in accuracy while maintaining robust AUC. These results confirm the importance of both input and score design.

**Impact of knowledge transfer strength.** Intuitively, the quality/strength of the distillation should impact the effectiveness of our approach. In Fig. 3, we show how our detection method depends on the strength of knowledge transfer during distillation. The parameter $\lambda$ controls the contribution of $\ell_{\text{soft}}$ in the distillation loss defined in Eq. (1), with higher $\lambda$ placing greater weights on matching the teacher. We use KL divergence and 1- ACS as the scores measuring the distance between the two models. We compute the scores between the outputs of the student model and either the teacher model or an independently trained model with the same architecture as the student. When $\lambda < 0.5$, the scores between the student and its teacher could be larger/comparable to those between the student and an independent model with the same architecture, and our detection rule will fail in such cases. As $\lambda$ increases, the student is more strongly regularized to align with the teacher, resulting in reliably smaller scores and improved detection performance.

# 6   Conclusion

We introduce the task of knowledge distillation detection that aims to identify whether a student model has been distilled from a given teacher from only the student's weights and the teacher's API access. Our model-agnostic framework combines data-free input synthesis with score maximization to make a prediction. The proposed method works on both classification and text-to-image generation. Experiments across diverse architectures and distillation methods show that our approach consistently outperforms strong baselines without requiring access to training data or teachers' weights. While detection may be less reliable when the student inherits only limited influence from the teacher, performance remains strong in typical distillation scenarios. Looking ahead, extending the framework to free-form inputs in diffusion and large language models represents a promising next step, supported by the model-agnostic nature of our approach.

## Acknowledgment

This project is supported in part by an NSF Award #2420724 and the Ross-Lynn Research Scholar Grant. Song's research is partially supported by the NVIDIA Academic Grant Program Award.

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

## Appendix

The appendix is organized as follows:

- In Sec. A1, we report additional quantitative results.
- In Sec. A2, we provide implementation details of our approach for reproducibility. The code will be released.
- In Sec. A3, we document the details of the compared baselines.
- In Sec. A4, we discuss broader impacts.

## A1  Additional Results

Table A1: Point-wise scores between the student models and each candidate models using KD method on CIFAR-10. The reported values are the mean ± standard deviation across runs.

| Teacher | Student | Ours (Method) | | | | MIA Filter + KL (Method) | | | |
|---|---|---|---|---|---|---|---|---|---|
| | | RN18 (Candidate) | DLA(Candidate) | DPN92(Candidate) | DN121(Candidate) | RN18(Candidate) | DLA(Candidate) | DPN92(Candidate) | DN121(Candidate) |
| RN18 | RN18 | 0.00 ± 0.00 | 0.27 ± 0.17 | 0.05 ± 0.02 | 0.05 ± 0.03 | 0.13 ± 0.04 | 0.21 ± 0.06 | 1.03 ± 0.19 | 0.13 ± 0.04 |
| | DLA | 0.05 ± 0.05 | 0.96 ± 0.13 | 0.76 ± 0.25 | 0.88 ± 0.29 | 0.11 ± 0.08 | 0.03 ± 0.03 | 2.97 ± 0.50 | 0.24 ± 0.19 |
| | DPN92 | 0.00 ± 0.00 | 0.13 ± 0.11 | 0.02 ± 0.04 | 0.00 ± 0.00 | 0.05 ± 0.02 | 0.06 ± 0.03 | 2.29 ± 0.44 | 0.14 ± 0.04 |
| | DN121 | 0.00 ± 0.00 | 0.28 ± 0.15 | 0.02 ± 0.01 | 0.02 ± 0.02 | 0.19 ± 0.10 | 0.03 ± 0.02 | 3.59 ± 0.42 | 0.22 ± 0.04 |
| DLA | RN18 | 0.01 ± 0.00 | 0.00 ± 0.00 | 0.09 ± 0.06 | 0.04 ± 0.02 | 0.12 ± 0.06 | 0.02 ± 0.01 | 2.19 ± 0.30 | 0.11 ± 0.03 |
| | DLA | 0.07 ± 0.03 | 0.00 ± 0.00 | 0.07 ± 0.04 | 0.06 ± 0.02 | 0.35 ± 0.14 | 0.02 ± 0.02 | 4.43 ± 0.45 | 0.45 ± 0.19 |
| | DPN92 | 0.02 ± 0.03 | 0.00 ± 0.00 | 0.00 ± 0.00 | 0.01 ± 0.00 | 0.11 ± 0.10 | 0.02 ± 0.00 | 2.15 ± 0.27 | 0.10 ± 0.04 |
| | DN121 | 0.19 ± 0.06 | 0.00 ± 0.00 | 0.33 ± 0.28 | 0.02 ± 0.03 | 0.15 ± 0.06 | 0.03 ± 0.01 | 2.23 ± 0.17 | 0.12 ± 0.07 |
| DPN92 | RN18 | 1.06 ± 0.21 | 2.16 ± 0.31 | 0.04 ± 0.05 | 0.64 ± 0.21 | 0.21 ± 0.03 | 0.31 ± 0.04 | 1.00 ± 0.07 | 0.21 ± 0.03 |
| | DLA | 1.30 ± 0.43 | 1.24 ± 0.22 | 0.00 ± 0.00 | 0.37 ± 0.14 | 0.38 ± 0.25 | 0.42 ± 0.24 | 1.14 ± 0.29 | 0.31 ± 0.16 |
| | DPN92 | 1.83 ± 0.19 | 1.66 ± 0.20 | 0.00 ± 0.00 | 0.28 ± 0.19 | 0.34 ± 0.08 | 0.46 ± 0.08 | 0.58 ± 0.07 | 0.32 ± 0.05 |
| | DN121 | 0.07 ± 0.09 | 0.19 ± 0.08 | 0.00 ± 0.01 | 0.03 ± 0.02 | 0.13 ± 0.06 | 0.05 ± 0.03 | 1.98 ± 0.36 | 0.07 ± 0.03 |
| DN121 | RN18 | 0.03 ± 0.01 | 0.14 ± 0.06 | 0.02 ± 0.02 | 0.00 ± 0.00 | 0.09 ± 0.05 | 0.06 ± 0.03 | 1.64 ± 0.15 | 0.05 ± 0.03 |
| | DLA | 0.03 ± 0.02 | 0.05 ± 0.04 | 1.02 ± 0.21 | 0.00 ± 0.00 | 0.12 ± 0.04 | 0.13 ± 0.03 | 1.27 ± 0.17 | 0.09 ± 0.07 |
| | DPN92 | 0.61 ± 0.28 | 0.19 ± 0.13 | 0.14 ± 0.20 | 0.00 ± 0.00 | 0.10 ± 0.05 | 0.11 ± 0.03 | 1.09 ± 0.13 | 0.05 ± 0.04 |
| | DN121 | 0.11 ± 0.07 | 0.17 ± 0.11 | 0.02 ± 0.02 | 0.00 ± 0.00 | 0.14 ± 0.06 | 0.17 ± 0.04 | 1.21 ± 0.17 | 0.10 ± 0.04 |

In Tab. A1, we compare the scores of our method and the best baseline across all student models on CIFAR-10. According to Tab. A1, we observe that some student models are easier to identify than others. For example, students distilled from RN18 are relatively easy cases, where both our method and the baseline achieve correct detection with large score gaps between the true teacher and other candidates. In contrast, harder cases such as RN18 students distilled from DN121 show much smaller differences, sometimes as low as 0.01 between the true teacher and the next closest candidate. Intuitively, the architectural similarity between the student and teacher models may influence the difficulty of detection.

Table A2: Point-wise scores between the student models and each candidate models using KD method. The reported values are the mean ± standard deviation across runs.

| Teacher | Student | Ours (Method) | | | MIA Filter + KL (Method) | | |
|---|---|---|---|---|---|---|---|
| | | RN18 (Candidate) | RN50 (Candidate) | RNXt50 (Candidate) | RN18 (Candidate) | RN50 (Candidate) | RNXt50 (Candidate) |
| RN18 | RN18 | 2.41 ± 1.07 | 7.01 ± 1.28 | 6.88 ± 1.18 | 0.13 ± 0.01 | 0.29 ± 0.02 | 0.70 ± 0.02 |
| | RN50 | 0.60 ± 0.40 | 5.12 ± 2.17 | 4.43 ± 1.01 | 0.15 ± 0.01 | 0.32 ± 0.01 | 0.72 ± 0.03 |
| | RNXt50 | 0.29 ± 0.25 | 1.90 ± 1.07 | 2.61 ± 0.95 | 0.18 ± 0.01 | 0.27 ± 0.01 | 0.68 ± 0.02 |
| RN50 | RN18 | 8.99 ± 1.22 | 9.45 ± 0.42 | 10.53 ± 1.47 | 0.36 ± 0.01 | 0.51 ± 0.04 | 0.92 ± 1.47 |
| | RN50 | 8.31 ± 2.19 | 5.48 ± 2.17 | 6.69 ± 2.08 | 0.26 ± 0.01 | 0.45 ± 0.02 | 0.79 ± 0.03 |
| | RNXt50 | 5.21 ± 1.72 | 3.13 ± 1.84 | 3.67 ± 1.47 | 0.34 ± 0.01 | 0.51 ± 0.03 | 0.78 ± 0.04 |
| RNXt50 | RN18 | 9.91 ± 1.33 | 11.22 ± 1.07 | 10.32 ± 1.26 | 0.45 ± 0.03 | 0.78 ± 0.05 | 1.01 ± 0.07 |
| | RN50 | 9.85 ± 1.48 | 10.45 ± 1.75 | 8.17 ± 0.95 | 0.36 ± 0.02 | 0.42 ± 0.02 | 0.76 ± 0.03 |
| | RNXt50 | 8.65 ± 1.69 | 9.39 ± 1.30 | 6.98 ± 1.98 | 0.52 ± 0.01 | 0.68 ± 0.02 | 0.82 ± 0.06 |

A similar conclusion can be drawn from Tab. A2 on ImageNet. For easy cases, such as students distilled from RN18, the true teacher can be accurately identified with large score gaps over other candidates. However, for harder cases, such as RN18 students distilled from RN50 or RNXt50, both our method and the baseline tend to incorrectly identify the teacher as RN18. These hard cases often involve significant architectural differences between the student and teacher models, suggesting that architectural mismatch contributes to detection difficulty.

In Tab. A3, we report the scores of our method and the best baseline across all student models on the text-to-image generation task. We use 1 - LPIPS in our method, which shows that the similarity

Table A3: Point-wise scores between the student models and each candidate models for text-to-image models. The reported values are the mean ± standard deviation across runs.

| Teacher | Student | Ours (Method) | | | GPT-2 + DINO (Method) | | |
|---------|---------|---------------|--|--|------------------------|--|--|
| | | SD-v2.1 (Candidate) | SDXL (Candidate) | PixArt (Candidate) | SD-v2.1 (Candidate) | SDXL (Candidate) | PixArt (Candidate) |
| SD-v2.1 | BK-SDM-v2-b | 0.69±0.01 | 0.28±0.00 | 0.19±0.00 | 1.00±0.00 | 0.98±0.00 | 0.97±0.00 |
| | BK-SDM-v2-s | 0.68±0.01 | 0.28±0.00 | 0.19±0.00 | 1.00±0.00 | 0.98±0.00 | 0.97±0.00 |
| | BK-SDM-v2-t | 0.70±0.01 | 0.27±0.00 | 0.18±0.00 | 1.00±0.00 | 0.98±0.00 | 0.97±0.00 |
| | AMD | 0.56±0.01 | 0.23±0.01 | 0.15±0.01 | 1.00±0.00 | 0.98±0.00 | 0.97±0.00 |
| SDXL | DMD2 | 0.24±0.01 | 0.32±0.01 | 0.20±0.00 | 0.98±0.00 | 0.99±0.00 | 0.98±0.00 |
| | SDXL-L-1 | 0.19±0.00 | 0.24±0.01 | 0.22±0.00 | 0.93±0.00 | 0.96±0.00 | 0.97±0.00 |
| | SDXL-L-2 | 0.25±0.01 | 0.41±0.01 | 0.18±0.00 | 0.99±0.00 | 0.98±0.00 | 0.98±0.00 |
| | SDXL-L-4 | 0.26±0.01 | 0.48±0.01 | 0.21±0.00 | 0.98±0.00 | 0.99±0.00 | 0.98±0.00 |
| | SDXL-L-8 | 0.23±0.01 | 0.55±0.01 | 0.21±0.00 | 0.97±0.00 | 0.99±0.00 | 0.98±0.00 |
| PixArt | AMD-PixArt | 0.19±0.00 | 0.37±0.00 | 0.43±0.00 | 0.98±0.00 | 0.98±0.00 | 0.99±0.00 |

between each student and their true teacher is the highest. For instance, BK-SDM-v2-b has $0.69\pm0.01$ to the teacher SD-v2.1, but much lower values to SDXL and PixArt. However, the baseline scores are nearly saturated across all candidates, typically 0.97–1.00 and making teacher identification infeasible. This saturation indicates a lack of discrimination in the embedding space used by the baseline.

## A2    Implementation details

**Teacher and student models in classification task.** We use ResNet-18, DLA, DPN92, and DenseNet121 as the backbone of the teacher and student models on CIFAR-10. We train the teacher/student models from scratch with these architectures first. Here we apply the same training strategy: stochastic gradient descent (SGD) with a learning rate of 0.01, momentum of 0.9, and weight decay of 5e-4; training length of 40 epochs with a batch size of 64. We apply the OneCycle learning rate scheduler, with a maximum learning rate of 0.1, computed over $45000/64$ steps per epoch and updated at every training step.

On ImageNet, since the teacher models are available on torchvision, we only need to train student models. The models are trained for 200 epochs with a batch size of 256. We use SGD with a learning rate of 0.1, momentum of 0.9, and weight decay of 1e-4. A cosine annealing learning rate scheduler is applied, with the minimum learning rate set to 0.0 and a total decay period over the whole epoch.

For the teacher model training, we use only the standard cross-entropy loss for classification. For student models trained with knowledge distillation (KD), only the KL divergence loss is used. In particular, for the RKD and OFA methods, we remove the hard-label supervision entirely and replace it with KL loss, so that the student models are trained solely under the guidance of the teacher.

In the RKD method, the total loss is a weighted combination of 0.3 KL loss and 0.7 RKD loss. Following the original RKD paper, we set the weight of the distance loss to 1 and the weight of the angle loss to 2. RKD loss is computed using the features after the final average pooling and flattening. To ensure dimensional compatibility between teacher and student models, we insert a fully connected projection layer for student models whose feature dimensions do not match those of the teacher. In the OFA method, the loss is a weighted combination of 0.3 KL loss and 0.7 OFA loss. For models with different architectures, we manually divide the hidden representations into four segments and select after-activation latents for computing the OFA loss.

**Input construction in classification task.** Our generator takes as input a noise vector and a class label, and produces a synthesized image conditioned on the label. Note that this label is first embedded using a learnable embedding layer. The normalized embedding is passed through a linear projection layer before being fed into the image generation pipeline.

On CIFAR-10, the projected embedding is transformed via a fully connected layer into a spatial feature map of shape $128 \times 8 \times 8$. This feature map is processed through three convolutional blocks, with bilinear upsampling applied between the blocks to restore spatial resolution:

- **Block 0:** A single BatchNorm layer.
- **Block 1:** A convolutional layer with 128 channels, followed by BatchNorm and LeakyReLU.

- **Block 2:** Two convolutional layers: one reducing to 64 channels and another producing the final image with the desired number of channels, followed by Tanh and BatchNorm.

On ImageNet, the input is also projected through a linear layer before being mapped to spatial features. The resulting vector is then passed through a fully connected layer and reshaped into a $128 \times 56 \times 56$ feature map. The convolutional backbone consists of three blocks, with bilinear upsampling applied between them:

- **Block 0:** Categorical Conditional BatchNorm (CCBN) is applied.
- **Block 1:** A convolutional layer with 128 filters, followed by CCBN and LeakyReLU activation.
- **Block 2:** One convolution with 128 channels that outputs the image channels, followed by CCBN, LeakyReLU. Followed by applying another convolutional layer, reducing to 64 channels, followed by a Tanh activation, and a final BatchNorm.

During training, the generator supports mixing noise latents and labels through weighted projection. The probability of applying mixup is set to 0.4. The number of inputs used for mixing is 2 for CIFAR-10 and 10 for ImageNet.

For CIFAR-10, we train for 400 epochs using a batch size of 200. The optimizer is Adam with $\beta_1 = 0.5$ and $\beta_2 = 0.999$. The label embedding dimension is 64. The learning rate is set to 0.001, using a multi-step schedule that decays by a factor of 0.1 at epochs 100, 200, and 300.

For ImageNet, we also train for 400 epochs with a batch size of 16. Adam is used with the same $\beta_1$ and $\beta_2$ values as in CIFAR-10. The label embedding dimension is set to 256 for ResNet18 students and 512 for ResNet50 and ResNeXt50 students. The learning rate is 0.001, with the same multi-step decay schedule.

**Hardware for experiments.** For training teacher and student models on CIFAR-10, we use a single A30 GPU. For ImageNet, student models are trained using two L40S GPUs.

In the first stage of our knowledge distillation detection pipeline, we train a generator to produce synthesized images in classification tasks using a single L40S GPU.

For score computation and prediction, we only perform inference with the student and teacher models. This stage uses a single A30 GPU for CIFAR-10 classification models, and a single L40S GPU for ImageNet classification and text-to-image generation models.

## A3 Baseline details

Most of the baselines have already been introduced in Sec. 5. Here, we formally describe MMD-FUSE in this section. MMD-FUSE is a kernel-based two-sample test. In our setting, the null and alternative hypotheses are:

$$\text{H}_0 : g_\theta \sim \mathcal{D}_{\text{distill}}(f) \quad vs. \quad \text{H}_1 : g_\theta \sim \mathcal{D}_{\text{indep}}, \tag{A15}$$

Let the student model outputs be denoted by $\{x_i\}_{i=1}^n$ and the teacher outputs by $\{y_i\}_{i=1}^n$. MMD-FUSE adaptively combines multiple kernels into a single test statistic. For a finite set of candidate kernels $\{k\}_{m=1}^M$, it computes each normalized MMD value and then takes a softmax over them:

$$\text{MMD\_FUSE} = \frac{1}{\beta} \log\left( \sum_{m=1}^M \exp(\beta \, \widehat{\text{MMD}}_m^2) \right),$$

where $\widehat{\text{MMD}}_m^2 = \frac{1}{n(n-1)} \sum_{i \neq j} k_m(x_i, x_j) + \frac{1}{n(n-1)} \sum_{i \neq j} k_m(y_i, y_j) - \frac{2}{n^2} \sum_{i=1}^n \sum_{j=1}^n k_m(x_i, y_j)$. To compute the p-value of the test, MMD-FUSE uses a permutation procedure. A smaller p-value indicates stronger evidence against the null hypothesis, suggesting that the student model is not a distilled version of the teacher.

In our experiment, we use two base kernels: the Laplace kernel and the RBF kernel. For each kernel, we select 10 different bandwidths. The number of permutations used for estimating the p-value is set to 1000, and the temperature scalar for the softmax aggregation is set to $\beta = 1$.

## A4   Broader impacts

This work contributes to model accountability by introducing a method for detecting whether a model has been trained via knowledge distillation from a specific teacher. This has positive implications for auditing model provenance, protecting intellectual property, and improving transparency in the deployment of machine learning systems. Our detection framework can help identify unauthorized reuse of proprietary models, especially in scenarios where model weights are publicly released but training procedures are not disclosed. At the same time, detection tools may influence how models are shared or reused, and like any statistical method, they may produce imperfect predictions. We suggest using such tools with awareness of their limitations and context.

