# OpenReview forum: "Knowledge Distillation Detection for Open-weights Models"
_NeurIPS.cc/2025/Conference — NeurIPS 2025 poster_

### Official Review · Reviewer_41Ct · 2025-07-03

**Clarity:** 3
**Significance:** 3
**Originality:** 2
**Rating:** 4
**Confidence:** 3

**Summary:**

This paper introduces the task of knowledge distillation detection, aiming to determine whether a student model has been distilled from a given teacher model, using only the student’s weights and the teacher’s as a black box (API). The authors propose a model-agnostic framework that combines data-free input synthesis, score computation, and decision-making stages to detect distillation, applicable to both classification and generative models. The method is tested on a variety of architectures for image classification and text-to-image generation, showing significant improvements in detection accuracy over existing baselines, with 59.6% better accuracy on CIFAR-10, 71.2% on ImageNet, and 20.0% for text-to-image generation.

**Questions:**

Address questions in weaknesses section.

**Ethical Concerns:**

["NO or VERY MINOR ethics concerns only"]

**Final Justification:**

Good paper, however current presentation is overselling the paper as a general method. Authors are recommended to improve the presentation and be specific where their method applies to.

**Limitations:**

Yes

**Paper Formatting Concerns:**

nil

**Quality:**

3

**Strengths And Weaknesses:**

Strengths:

Novel Task and Relevance: The paper addresses a timely and critical issue of knowledge distillation detection, which is becoming more important with concerns about intellectual property, and unauthorized replication. The problem setting where the teacher is closed sources used to distill a smaller open sourced model is an interesting problem. The proposed approach is adaptable and can be used for a wide variety of models (both classification and generative), making it highly versatile and applicable across different domains. The method is validated on multiple datasets and architectures, including CIFAR-10, ImageNet, and text-to-image generation tasks. The results show impressive improvements over strong baselines. The paper is also generally well written an easy to follow.

Weaknesses

The paper discusses knowledge distillation which alignment between the teacher and student models. However, closed-source teacher models only have API access, limiting the ability to directly access and align the internal representations between the teacher and student. Meaning with these closed source model, the only knowledge distillation method one can use is to generate outputs from the teacher and use it to train a student. Does the paper proposed techniques also address this scenario?

The paper's approach assumes a scenario where a fixed set of K teacher models is provided, and the goal is to identify which teacher was used to distill the student model. However, this assumption does not address the more general and arguably more important problem: determining whether a given student model was distilled from a given teacher model or not. In real-world applications, the number of possible teacher models could be enormous, if not infinite, making it difficult to define such a finite subset of potential teachers.  A key test for your setup could involve running the proposed method with a set of K models, where none of them have been involved in the distillation of the student model. The interesting question here would be whether the method falsely identifies any of those models as the source, potentially leading to false positives?

Does the proposed method apply to large language models (LLMs) ? Given that the paper primarily focuses on image classification and text-to-image generation tasks, it would be interesting to know if the detection method can also be generalized to handle LLMs. I recommend an experiment on this. This could be a simple sentiment analysis case (also see [1]).

I recommend you summarize your method and approach in an algorithm environment for better clarity.

[1] Less is More: Task-aware Layer-wise Distillation for Language Model Compression https://arxiv.org/abs/2210.01351

---

> ### Author Rebuttal · Authors · 2025-07-30
>
> Thank you for the thoughtful and encouraging review. We appreciate the recognition on the novelty, relevance, and writing quality of this work. We now address individual questions.
>
> > Q18. Senario where distillation can only use closed-source teacher model's output?
>
> To clarify, the closed-source assumption applies to the detection phase and not to the distillation process itself. Our detection method does not distinguish between *how* a student model has been distilled. For example, one of the experiment uses the original knowledge-distillation (KD) to train the student model, which only uses a closed-source teacher model's output.
>
>
>
> > Q19. Motivation vs experiment setup.
>
> Thanks for pointing this out. We agree that we should make the connection between the motivation and the task formulation more explicit. Below, we carefully discuss the implications, clarify the setup, and how we will update the manuscript.
>
> 1. **Motivated:** Given a model, predict whether it has been distilled **FROM** a teacher model. (A teacher model candidate is required)
> 2. **Formulated:** Given a model, predict **WHICH** of the teacher model has been used to distill the student model. (A *set* of teacher candidates is required \& assuming the student model has been distilled from one of them)
>
> In the introduction, we motivated with (1) but presented a task formulation of (2). We now recognize that this connection may not have been sufficiently emphasized.
>
>
> We formulated the task as a multiple choice problem as it is straightforward and with results easy to understand. The main difference between (1) and (2) is that (1) further requires an absolute threshold for decision making. In contrast, the multiple choice setting performs a relative comparison across a set of candidate teacher models. This allows us to select the most likely teacher without requiring an explicit threshold. Our method can be modified from (2) to (1), see Q20.
>
>
> > Q20. Extending to the determining whether a given student model was distilled from a given teacher model or not.
>
> Under our distillation detection framework, transferring from (2) to (1) would require: **(A)**  a calibrated score function and **(B)** a suitable decision rule. As an example, we can define the score as the p-value from a statistical independence test between the teacher and student. The decision of (1) can be made by checking whether this p-value falls below a chosen significance level, e.g., 0.05.
>
>
> Next, we implement the HSIC test [A] to obtain p-value as score for the text-to-image generation task with 3 teacher models and 10 student models. Each of the student model is paired with its true teacher (positive) and two random candidate teachers (negative). A pair is predicted as distillation if the p-value is below 0.05. From the 30 teacher-student pairs, accuracy and F1 score are computed from these binary predictions, and AUC is based on the scores across different thresholds for p-value. The results are provided in Table R6. These results show that generalizing to the formulation (1) is possible. We view this as a natural extension and a valuable direction for future work.
>
>
> Table R6. Performance of distillation detection using the HSIC test on the text-to-image task.
> | Input size | Acc. | AUC | F1 |
> | -------- | -------- | -------- | -------- |
> | 10     | 0.74  $\pm$ 0.05     | 0.76 $\pm$ 0.07 | 0.43 $\pm$ 0.09 |
> | 50     | 0.86  $\pm$ 0.04     | 0.91 $\pm$ 0.07 | 0.80 $\pm$ 0.06 |
> | 100    | 0.80 $\pm$ 0.04      | 0.94 $\pm$ 0.04 | 0.75 $\pm$ 0.05 |
>
> We will update the abstract and introduction of the manuscript to clarify the connection between the motivation and the formulation.
>
> [A] Gretton, Arthur, et al. "A kernel statistical test of independence." In Proc. NeurIPS 2007.
>
>
> > Q21. Does the proposed method apply to LLMs?
>
> In theory, our framework an be adapted to LLMs with specific choices for synthesis and scoring required modification, as they are domain-specific (NLP). For synthesis, inspiration can be drawn from "Data-Free Quantization for LLMs" [B]. For the scoring function, we could consider utilizing text embedding similarity, which converts the discrete output back into a continuous domain.
>
> [B] Liu, Zechun, et al. "LLM-QAT: Data-free quantization aware training for large language models." arXiv preprint arXiv:2305.17888 (2023).
>
> We will add this discussion to the limitations section and clarify the scope of this paper. While we attempted to be clear by mentioning “image classification and text-to-image generation” in the abstract, we will further clarify in the writing. For instance, would including the term "Vision Models" in the title be helpful?
>
>
>
> > Q22. Summarize method and approach in an algorithm.
>
> Thanks for this great suggestion. We only provided a visual overview in Fig. 1. We agree that presenting the method in an algorithm would be easier to follow. We will modify.

---

> > ### Comment · Reviewer_41Ct · 2025-08-09
> >
> > Thank you for your detailed rebuttal. These discussions should be included in the revised manuscript (Q18-Q20).
> >
> > Regarding (Q21), I don't think it was clear by adding "image classification and text-to-image generation" that your setting were for image models (method seems general initially). Especially with words like "model-agnostic" etc. I would recommend clarifying that early in the abstract that this is for images. The title should also be updated to be more specific,  "Open-weights Models" is a very large set of model types.
> >
> > I have updated my score conditioned these edit will be made in the revised version.

---

> > > ### Author Response · Authors · 2025-08-09
> > >
> > > Dear Reviewer 41Ct,
> > >
> > > Thank you for the constructive feedback and for updating your score. We will incorporate the discussions from Q18–Q20 into the revised manuscript, make it clear in the abstract, introduction, and method sections that our setting focuses on image models, and update the title to be more specific as suggested.
> > >
> > > Best regards,
> > > Authors

---

### Official Review · Reviewer_4543 · 2025-07-03

**Clarity:** 3
**Significance:** 1
**Originality:** 2
**Rating:** 3
**Confidence:** 3

**Summary:**

This paper presents a method for determining which teacher model distilled a student model when only the student model and teacher API are available.

**Questions:**

I believe the primary issue with this paper is the lack of value in the research problem it addresses. If the authors can demonstrate the practical significance of this research setting, I would reconsider my rating.

**Ethical Concerns:**

["NO or VERY MINOR ethics concerns only"]

**Final Justification:**

The author's response has largely addressed my concerns regarding the application scenario. However, I still believe that merely selecting the most likely teacher among a few does not hold much significance. Determining whether distillation is present or not is far more meaningful. Therefore, I will raise my rating to borderline reject.

**Limitations:**

Please refer to the question section.

**Paper Formatting Concerns:**

No paper formatting concerns.

**Quality:**

2

**Strengths And Weaknesses:**

Strengths:

1. This paper is straightforward and easy to understand. In terms of results, it shows a significant improvement over the baseline methods.

Weaknesses:

1. My primary concern is the value of the problem being addressed. This paper investigates a scenario under very stringent conditions—having only the teacher's API and the student's model, without access to the data used for distillation. Such situations are rare and lack clear application contexts. Additionally, the method proposed in the paper can only identify the most likely teacher from multiple options, but it cannot determine whether distillation was used on the student at all. The latter capability would have greater practical value.

2. The main contribution of this paper is the proposal of score functions and input data generation methods for various scenarios, along with experimental validation. However, these methods are straightforward and lack novelty.

---

> ### Author Rebuttal · Authors · 2025-07-30
>
> We thank the reviewer for the feedback. We address individual questions below. We believe there are some misunderstanding and look forward to clarifying them.
>
>
> > Q14. Novelty and contribution.
>
> To the best of our knowledge, the introduced task of distillation detection has not been addressed in prior research. We present a framework that breaks down the distillation detection process into three main components: input construction, score computation, and decision-making rules. We demonstrate the framework’s effectiveness in image classification and text-to-image tasks, comparing it against several different baselines by adapting related areas. We believe that our work enhances the understanding of distillation detection beyond the  current literature and has real-world implications, as discussed in Q16.
>
> > Q15. The paper investigates a scenario under very stringent conditions.
>
> We understand the reviewer’s concern, but we believe the setting we study is realistic and practical. In many real-world cases, especially with commercial models, the training data is not available, and the teacher model is only accessible via an API. While the setting may seem strict, this is also what will make our method effective in real-world circumstances.
>
> Please note that we have included results for a more relaxed setup, where training data is accessible (as shown in the Oracle case in Tables 1 and 2), and our method still performs well. This shows that the approach remains effective across different assumptions.
>
> > Q16. Real world application for distillation detection.
>
> Besides the scientific question, we foresee that distillation detection will have the following real-world use cases:
> 1. **Detection terms of service violation:** In the terms of service for OpenAPI services (OpenAI, Gemini, Midjourney, etc.), it is common to have some form of clause that prohibits users from training/distilling the services' output. The ability to detect distillation will enable the detection of such a violation.
> 2. **Tracing content origin:** Recently, there has been an interest in determining whether a model has been trained on data with IP protection. This would ensure proper accounting and crediting of the data source. Distillation detection will be a useful tool to trace these data sources across models. This closes the loophole that one could claim that they never trained on certain data, where they "distilled" from a model that did.
>
> We believe our work is an important first step towards these directions. We will clarify and provide a more in-depth discussion in the introduction.
>
>
> > Q17. Most likely teacher vs. whether distillation occur.
>
> Thanks for pointing this out. We agree that we should make the connection between the motivation and the task formulation more explicit. Below, we carefully discuss the implications, clarify the setup, and how we will update the manuscript.
>
> 1. **Motivated:** Given a model, predict whether it has been distilled **FROM** a teacher model. (A teacher model candidate is required)
> 2. **Formulated:** Given a model, predict **WHICH** of the teacher model has been used to distill the student model. (A *set* of teacher candidates is required \& assuming the student model has been distilled from one of them)
>
> In the introduction, we motivated with (1) but presented a task formulation of (2). We now recognize that this connection may not have been sufficiently emphasized.
>
>
> We formulated the task as a multiple choice problem as it is straightforward and with results easy to understand. The main difference between (1) and (2) is that (1) further requires an absolute threshold for decision making. In contrast, the multiple choice setting performs a relative comparison across a set of candidate teacher models. This allows us to select the most likely teacher without requiring an explicit threshold.
>
> Under our distillation detection framework, transferring from (2) to (1) would require: **(A)**  a calibrated score function and **(B)** a suitable decision rule. As an example, we can define the score as the p-value from a statistical independence test between the teacher and student. The decision of (1) can be made by checking whether this p-value falls below a chosen significance level, e.g., 0.05.
>
>
> Next, we implement the HSIC test [A] to obtain p-value as score for the text-to-image generation task with 3 teacher models and 10 student models. Each of the student model is paired with its true teacher (positive) and two random candidate teachers (negative). A pair is predicted as distillation if the p-value is below 0.05. From the 30 teacher-student pairs, accuracy and F1 score are computed from these binary predictions, and AUC is based on the scores across different thresholds for p-value. The results are provided in Table R5. These results show that generalizing to the formulation (1) is possible. We view this as a natural extension and a valuable direction for future work.
>
>
> Table R5. Performance of distillation detection using the HSIC test on the text-to-image task.
> | Input size | Acc. | AUC | F1 |
> | -------- | -------- | -------- | -------- |
> | 10     | 0.74  $\pm$ 0.05     | 0.76 $\pm$ 0.07 | 0.43 $\pm$ 0.09 |
> | 50     | 0.86  $\pm$ 0.04     | 0.91 $\pm$ 0.07 | 0.80 $\pm$ 0.06 |
> | 100    | 0.80 $\pm$ 0.04      | 0.94 $\pm$ 0.04 | 0.75 $\pm$ 0.05 |
>
> We will update the abstract and introduction of the manuscript to clarify the connection between the motivation and the formulation.
>
> [A] Gretton, Arthur, et al. "A kernel statistical test of independence." In Proc. NeurIPS 2007.

---

### Official Review · Reviewer_bDcj · 2025-07-03

**Clarity:** 4
**Significance:** 3
**Originality:** 4
**Rating:** 5
**Confidence:** 3

**Summary:**

This paper introduces a means to determine if a student model was distilled from a specific teacher model, using the student's weights and the teacher's API. Provenance is an important and thorny problem, and unlikely many proposed problems in ML security, the assumptions around model & API access appear to be practical and not contrived. The method itself works by generating synthetic inputs that are predicted confidently with the student model, these are then scoring under the model and under the candidate teacher models, to find the pairing of student and teacher that are most aligned in their predictions. Overall this looks well motivated, and the approach is demonstrated to work empirically via experiments on image classification and text to image generation.

**Questions:**

1. Could you adapt the method to work purely from API access for student and candidate teacher models? This would complicate the synthesis step, but would the other parts of the method be applicable?
1. Given this method now exists, could an adversary use this to "hide" their dependence on a specific teacher model? This might work by ensuring the confidence of the model is high on out-of-domain instances, or adding label noise to training, or other means. Might it be possible to reach similar model quality, despite obscuring the provenance?
1. As well as intentional distillation, there's unintentional copying of another model, e.g., by training on web crawls that include outputs from other models. This is a big problem for modern foundation models, in that it can be very hard to identify, and can lead to performance, legal and other issues. This would be an interesting topic to discuss in the introduction or discussion parts of the paper.
1. A related problem is when a model is trained on the same private dataset. Wouldn't this have a similar fingerprint on the model activations as distillation? Could this be differentiated from distillation?
1. Confident predictions from the model are as a signal for data synthesis. Would this be a strong heuristic where the predictions are incorrect. That is, agreeing on a mistake with the teacher is much more informative than agreeing on a correct answer.
1. It's one thing to predict a score for each teacher, but can this be cast as a likelihood that the models were trained independently, and thus used in a statistical test of independence. This would help in legal application of the method.
1. Can you comment on how the SVD method in 4.2 will scale for very multiclass scenarios, with many 1000s of labels and potentially sparse activations.

**Ethical Concerns:**

["NO or VERY MINOR ethics concerns only"]

**Final Justification:**

The authors gave a detailed response, resolving the questions I had. I stand by my strong positive assessment of the work.

**Limitations:**

yes

**Quality:**

4

**Strengths And Weaknesses:**

Strengths:

1.  **Timely problem:** The paper tackles the growing concern of model provenance and unauthorized replication, something that currently is not well understood.
2.  **Methodology:** The method of knowledge distillation detection is well motivated, model-agnostic, and is shown to work for both classification and generative models, with only limited access to the models themselves (but see more later).
3.  **Good results:** They show solid improvements in detection accuracy over strong baselines in image classification (CIFAR-10, ImageNet) and text-to-image generation tasks.

Weaknesses:

1.  **Limited application to text inputs/outputs:** The paper focuses on image-based tasks (classification and text-to-image generation). While text-to-image generation involves text prompts as input, the "empty string" approach simplifies the input generation. It is not clear how the proposed data-free input synthesis and scoring mechanisms would directly translate or perform in entirely discrete domains like text-to-text models (e.g., detecting distillation for large language models).
2.  **Reliability when student inherits limited influence from teacher:** The authors acknowledge that "detection may be less reliable when the student inherits only limited influence from the teacher" as shown in the ablation in Fig. 3. I think in most practical scenarios the student will be trained to replicate the teacher closely, so this may not be a big issue.
3.  **Computational costs:** The cost of being data-free is the need to synthesize inputs. This doesn't seem very scalable. There's some discussion in the appendix of the costs of synthesis, but I'd appreciate this being discussed in the main paper.

---

> ### Author Rebuttal · Authors · 2025-07-30
>
> We thank the reviewer for recognizing our timely problem, model‑agnostic approach, and strong performance on both classification and text‑to‑image tasks.
>
>
> > Q4. Generalization to text-to-text models
>
> The high-level detection framework, of data-free synthesis followed by score computation, applies to text-to-text models. However, the specific choices for synthesis and scoring required modification, as they are domain-specific (NLP). For synthesis, inspiration can be drawn from "Data-Free Quantization for LLMs" [B]. For the scoring function, we could consider utilizing text embedding similarity, which converts the discrete output back into a continuous domain.
>
> [B] Liu, Zechun, et al. "LLM-QAT: Data-free quantization aware training for large language models." arXiv preprint arXiv:2305.17888 (2023).
>
> We will add this discussion to the limitations section and clarify the scope of this paper. While we attempted to be clear by mentioning “image classification and text-to-image generation” in the abstract, we will further clarify in the writing. For instance, would including the term "Vision Models" in the title be helpful?
>
>
>
> > Q5. Reliability when student inherits limited influence from teacher is not a big issue.
>
> Thanks for this comment. We will incorporate this view and defer these analysis to the appendix.
>
> > Q6. Computational costs
>
> Thank you for the suggestion. As suggested, we will move the computational cost from the appendix to the main paper.
>
>
> > Q7. Adapt the method to work purely from API access for both student/teacher models?
>
> In some context, it is possible to only require API access for both student and teacher models.
>
> - For the text-to-image setting, our current experiments already operate under API-only access to the student model. As we use the "empty string" approach to data generation, and the score function does not require model weights.
>
> - For the classification case, the current synthesis step depends on backpropagation through the student, which is not feasible with only API access. One possible direction is to explore black-box gradient approximation methods [C], however, such exploration is beyond the scope of this work.
>
> [C] Grathwohl, Will, et al. "Backpropagation through the void: Optimizing control variates for black-box gradient estimation." Proc. ICLR, 2018.
>
>
> > Q8. Could an adversary use this to "hide" their dependence on a specific teacher model?
>
> This is a possible scenario. Currently, our experiment is based on existing distillation methods. An adversary could try to modify their distillation approach to evade detection, e.g., deliberating injecting noise during the process.  This adversarial setting is an interesting future direction. We will add a discussion.
>
>
>
> > Q9. Discussion of unintentional copying of another model.
>
> Thanks for this comprehensive point of view. Currently, this work focus on the intentional distillation to establish a concrete framework and setup. We agree that unintentional copying is relevant, yet out of the scope of this work. We will discuss unintentional copying as a limitations/future work.
>
>
>
> > Q10. Cases when models are trained on the same private dataset.
>
> Thanks for bringing up this interesting experiment. Here we provide experiment results in Table R2 for image classification, where we compare the KL distance between (a) the teacher and the distilled student and (b) the teacher and an independently trained model.
>
> These models are trained on CIFAR10 with the corresponding architecture. We can observe the KL distance of (a) is consistently smaller than (b). This means that our approach can differentiate distillation from simply training from the same dataset.
>
> **Table R2: Results from training on the same private dataset.**
> | Architecture | Distilled | Independent |
> | -------- | -------- | -------- |
> | ResNet-18 | $3.13e^{-4}$| $5.22e^{-1}$ |
> |DLA |$1.89e^{-4}$| $5.85e^{-1}$ |
> |DPN-92 |$3.31e^{-4}$| $1.64e^{-1}$ |
> |DenseNet-121 |$9.71e^{-5}$| $1.47e^{-1}$ |
>
>
> > Q11. Would agreement on incorrect predictions serve as a stronger signal for distillation?
>
> In our data synthesis process, with its data-free nature, we don't have the ground truth label. As such, it is unclear how to define a “mistake”.
>
> To test the reviewer's hypothesis, we provide experiment results by assuming we have access to the real training data. Specifically, for each candidate teacher, we count how often the student and teacher agree on incorrect predictions, and we select the teacher with the highest count. The results are summarized in Table R3. We can indeed see an improvement by using agreement on mistakes as the score and decision rule.
>
> Table R3. Accuracy when using "agreement on mistakes".
> | Input size | Acc. (agree on mistakes)  | Acc. (Oracle) |
> | -------- | -------- | -------- |
> | 10  | 0.71 $\pm$ 0.06 | 0.67 $\pm$ 0.07 |
> | 50  | 0.97 $\pm$ 0.05 | 0.87 $\pm$ 0.04 |
> | 100 | 1.00 $\pm$ 0.00 | 0.95 $\pm$ 0.02 |
>
> We will update the paper to include these findings, and we will also add a discussion of alternative heuristics, including agreement on mistakes, as potential extensions under more relaxed assumptions.
>
>
>
> > Q12. Can the method estimate a likelihood that the student and a given teacher were trained independently?
>
> Yes, we provide additional results of using independence test on the task of text-to-image generation in Table R4. Here we implement the HSIC test [A] to obtain p-value as the score with 3 teacher models and 10 student models. Each of the student model is paired with its true teacher (positive) and two random candidate teachers (negative). A pair is predicted as distillation if the p-value is below 0.05. From the 30 teacher-student pairs, accuracy and F1 score are computed from these binary predictions, and AUC is based on the scores across different thresholds for p-value. We can see our framework can be adapt to using p-values as the score.
>
> Table R4. Performance of distillation detection using the HSIC test on the text-to-image task.
> | Input size | Acc. | AUC | F1 |
> | -------- | -------- | -------- | -------- |
> | 10     | 0.74  $\pm$ 0.05     | 0.76 $\pm$ 0.07 | 0.43 $\pm$ 0.09 |
> | 50     | 0.86  $\pm$ 0.04     | 0.91 $\pm$ 0.07 | 0.80 $\pm$ 0.06 |
> | 100    | 0.80 $\pm$ 0.04      | 0.94 $\pm$ 0.04 | 0.75 $\pm$ 0.05 |
>
>
> [A] Gretton, Arthur, et al. "A kernel statistical test of independence." In Proc. NeurIPS 2007.
>
>
> > Q13. Comment on how the SVD method in 4.2 will scale for very multiclass scenarios
>
> The SVD step in Sec. 4.2 has a complexity of $O(C^3)$ for $C$ classes. Note that it is computed only once per teacher-student pair as part of the set-level score. To make the computation efficient for large $C$,  SVD approximations such as truncated or sparse SVD [D] can applied.
>
> [D] Yang D, Ma Z, Buja A. A sparse singular value decomposition method for high-dimensional data. Journal of Computational and Graphical Statistics, 2014

---

> > ### Comment · Reviewer_bDcj · 2025-08-06
> >
> > Thank you for the very thorough response to all my questions, including new results which will help to improve the paper. I stand by my strong positive assessment of the work.

---

> > > ### Author Response · Authors · 2025-08-09
> > >
> > > Dear Reviewer bDcj,
> > >
> > > Thank you for your encouraging feedback and strong positive assessment. We are glad that the additional results and clarifications were helpful, and we will add these improvements to the revised manuscript.
> > >
> > > Best regards,
> > > Authors

---

### Official Review · Reviewer_nZTZ · 2025-07-06

**Clarity:** 4
**Significance:** 3
**Originality:** 3
**Rating:** 5
**Confidence:** 3

**Summary:**

This paper proposes and addresses the task of "knowledge distillation detection", wherein the goal is to determine whether an existing model has been distilled from a separate teacher model - given only access to student model weights and the teacher API. The final proposed mechanisms to tackle this problem comprises three primary stages: (1) probing model behaviour via synthetically generated data, (2) extracting statistical scores from model responses, and (3) applying a given decision rule to determine whether distillation has occurred or not - all without making particular requirements on the model architectures of choice.
Experiments are conducted on smaller image classification and generation scenarios (e.g. ResNet-style models, on CIFAR-10/ImageNet, and models such as SD-v2.1 / SDXL) show high efficacy of the proposed mechanism.

**Questions:**

I do think this paper is a relevant contribution - it tries to formalize an important research problem (detecting whether distillation has occured), and within the setting tested does so really well. Unfortunately, I am somewhat struggling to make the connection between the specific research problem and the realistic  scenarios within which distillation detection would occur. It would be great if the authors could provide context here.

**Ethical Concerns:**

["NO or VERY MINOR ethics concerns only"]

**Final Justification:**

The authors have done a great job clarifying the mismatch between formulation & motivation, alongside some initial experimental support that one can transition into the other. If the authors can work this into the final version of the submission, I'm happy to recommend acceptance.

**Limitations:**

Not specifically addressed.

**Quality:**

3

**Strengths And Weaknesses:**

__Strengths:__
* The paper is well written, organised and easy to follow - motivating the task well, before diving into their proposed first approach to tackle this problem.
* To the best of my knowledge, the proposed distillation detection task, the problem formulation, and the suggested approach to test for distillation is novel.
* The experiments conducted are extensive, and convincing; showcasing convincing results across baseline methods across both classification distillation detection (Tab 1 and 2), as well as for T2I models in Tab. 3

__Weaknesses:__

My main issues lies in the discrepancy between the formulated problem setting and the primary motivation of the paper / distillation detection issue: instead of detecting >whether a model was distilled from another model<, the task is that of detecting which model was distilled from - which excludes the difficult scenario of determining whether any distillation has actually happened. It would be great if the authors could provide insights here; whether the proposed method could be extended to this much more realistic and general scenario, and clearly distinguish the importance of the actually tested scenario of determining which model was distilled from. In a legal issue scenario where someone was accused of distilling e.g. from some proprietary model, just being able to say that there is a higher chance of distillation w.r.t. To some generic baseline would hold value.

---

> ### Author Rebuttal · Authors · 2025-07-30
>
> We thank the reviewer for the recognition of our clear writing, novel problem formulation, and strong experimental results.
>
> > Q1. Formulated problem setting vs. the primary motivation.
>
> Thank you for pointing this out. We agree that we should make the connection between the motivation and the task formulation more explicit. Below, we carefully discuss the implications, clarify the setup, and how we will update the manuscript.
>
> 1. **Motivated:** Given a model, predict whether it has been distilled **FROM** a teacher model. (A teacher model candidate is required)
> 2. **Formulated:** Given a model, predict **WHICH** of the teacher model has been used to distill the student model. (A *set* of teacher candidates is required \& assuming the student model has been distilled from one of them)
>
> In the introduction, we motivated with (1) but presented a task formulation of (2). We now recognize that this connection may not have been sufficiently emphasized.
>
>
> We formulated the task as a multiple choice problem as it is straightforward and with results easy to understand. The main difference between (1) and (2) is that (1) further requires an absolute threshold for decision making. In contrast, the multiple choice setting performs a relative comparison across a set of candidate teacher models. This allows us to select the most likely teacher without requiring an explicit threshold. Our method can be modified from (2) to (1), see Q2.
>
> > Q2. Extending to the scenario of the motivated problem.
>
> Under our distillation detection framework, transfering from (2) to (1) would require: **(A)**  a calibrated score function and **(B)** a suitable decision rule. As an example, we can define the score as the p-value from a statistical independence test between the teacher and student. The decision of (1) can be made by checking whether this p-value falls below a chosen significance level, e.g., 0.05.
>
>
> Next, we implement the HSIC test [A] to obtain p-value as score for the text-to-image generation task with 3 teacher models and 10 student models. Each of the student model is paired with its true teacher (positive) and two random candidate teachers (negative). A pair is predicted as distillation if the p-value is below 0.05. From the 30 teacher-student pairs, accuracy and F1 score are computed from these binary predictions, and AUC is based on the scores across different thresholds for p-value. The results are provided in Table R1. These results show that generalizing to the formulation (1) is possible. We view this as a natural extension and a valuable direction for future work.
>
>
> Table R1. Performance of distillation detection using the HSIC test on the text-to-image task.
> | Input size | Acc. | AUC | F1 |
> | -------- | -------- | -------- | -------- |
> | 10     | 0.74  $\pm$ 0.05     | 0.76 $\pm$ 0.07 | 0.43 $\pm$ 0.09 |
> | 50     | 0.86  $\pm$ 0.04     | 0.91 $\pm$ 0.07 | 0.80 $\pm$ 0.06 |
> | 100    | 0.80 $\pm$ 0.04      | 0.94 $\pm$ 0.04 | 0.75 $\pm$ 0.05 |
>
> We will update the abstract and introduction of the manuscript to clarify the connection between the motivation and the formulation.
>
> [A] Gretton, Arthur, et al. "A kernel statistical test of independence." In Proc. NeurIPS 2007.
>
>
> > Q3. Real world use cases for distillation detection.
>
>
> Besides the scientific question, we foresee that distillation detection will have the following real-world use cases:
> 1. **Detection terms of service violation:** In the terms of service for OpenAPI services (OpenAI, Gemini, Midjourney, etc.), it is common to have some form of clause that prohibits users from training/distilling the services' output. The ability to detect distillation will enable the detection of such a violation.
> 2. **Tracing content origin:** Recently, there has been an interest in determining whether a model has been trained on data with IP protection. This would ensure proper accounting and crediting of the data source. Distillation detection will be a useful tool to trace these data sources across models. This closes the loophole that one could claim that they never trained on certain data, where they "distilled" from a model that did.
>
> We believe our work is an important first step towards these directions. We will clarify and provide a more in-depth discussion in the introduction.

---

> > ### Comment · Reviewer_nZTZ · 2025-08-05
> > **Response to Rebuttal**
> >
> > The authors did a really great job clarifying the motivation / formulation mismatch, including the addition, alongside initial thresholding experiments to transition from the existing formulation to the generalized motivation. Would be great to see these experiments in the final version of the paper.
> >
> > Consequently, I'm happy to continue advocating for acceptance!

---

> > > ### Author Response · Authors · 2025-08-06
> > >
> > > We appreciate the reviewer's feedback! All results and clarifications will be included in the paper. Thank you for the constructive suggestions, which helped to improve this work.

---

### Author Response · Authors · 2025-08-06

We would like to thank the reviewers and AC for their time and thoughtful feedback. In our rebuttal, we have addressed the main concerns and provided additional experiments and clarification to strengthen this work. We look forward to answering any further comments.

---

### Note · Authors · 2025-08-12

We thank the reviewers and AC for their constructive feedback. We propose the novel task of knowledge distillation detection under a realistic setting where only the student’s weights and the teacher’s API are available. We introduce a model-agnostic framework that combines data-free input synthesis, score computation, and decision rules. We show the application of our approach to both image classification and text-to-image generation.

We appreciate that three reviewers found the problem timely, the methodology well-motivated, and the results convincing, with three reviewers expressing a positive view of our work after the discussion. We did not hear back from Reviewer 4543, but we believe that all concerns have been addressed.


To summarize, we addressed the following:

- **Motivation vs. formulation**: Multiple reviewers asked about the difference between identifying the most likely teacher (our current setting) and determining whether a given student model was distilled from a given teacher model or not. We clarified that our framework naturally extends to the latter by calibrating scores and applying statistical decision rules, and we provided additional experiments using HSIC tests to demonstrate feasibility. The details are in our responses to Q1-2, Q17, and Q19-20 in the rebuttal.
- **Applicability and scope**: We explained that while our paper focuses on vision models, the framework can be adapted to other domains, such as LLMs, with domain-specific synthesis and scoring methods. The details are in our responses to Q4 and Q21 in the rebuttal.
- **Practicality and real-world value**: We discussed realistic use cases, including detecting violations of terms of service and tracing content origin to ensure intellectual property compliance. The details are in our responses to Q3 and Q16 in the rebuttal.
- **Additional technical concerns**: We also addressed various implementation, scalability, and robustness questions, as detailed in our responses to Q6–13 in the rebuttal.


The following changes will be made following reviewers' suggestions:

- Explicitly connect the motivation and task formulation in the abstract, introduction, and method.
- Clarify the focus on image models in the title and early sections.
- Incorporate all additional experiments and discussions from the rebuttal phase.

We believe this work is an important first step towards distillation detection and would be valuable to the community.

---

### Decision · Program_Chairs · 2025-09-17

**Decision:**

Accept (poster)

**Comment:**

The paper presents an approach to the problem of knowledge distillation detection, which is a timely and relevant issue in the field of artificial intelligence, particularly concerning model provenance and unauthorized replication. The reviewers generally praise the paper for its well-organized content, clarity, and the novelty of knowledge distillation detection problem. The methodology is also appreciated for its experiments in image classification (CIFAR-10, ImageNet) and text-to-image generation scenarios. Meanwhile, some constructive criticisms and suggestions for improvement are also provided, including the problem settings, the scope of detection capabilities and the computational costs.

Given the balance of strengths and weaknesses, and considering the critical importance of the topic, I decide to accept the work with strong encouragement for the authors to undertake significant revisions to address the reviewers' concerns. The novelty and relevance of the work are clear, but realizing its full potential and ensuring broader applicability require addressing the identified weaknesses.